# ATR promotes mTORC1 activity via de novo cholesterol synthesis

Naveen Kumar Tangudu[1,2], Alexandra N Grumet [ID][3], Richard Fang[1,2], Raquel Buj [ID][1,2], Aidan R Cole [ID][1,2], Apoorva Uboveja[1,2], Amandine Amalric[1,2], Baixue Yang[1,2,4], Zhentai Huang[1,2], Cassandra Happe[5], Mai Sun[5], Stacy L Gelhaus [ID][1,5], Matthew L MacDonald[5,6], Nadine Hempel [ID][2,7], Nathaniel W Snyder [ID][8], Katarzyna M Kedziora[9], Alexander J Valvezan [ID][3✉] & Katherine M Aird [ID][1,2✉]

## Abstract

DNA damage and cellular metabolism exhibit a complex interplay characterized by bidirectional feedback. Key mediators of these pathways include ATR and mTORC1, respectively. Previous studies established ATR as a regulatory upstream factor of mTORC1 during replication stress; however, the precise mechanisms remain poorly defined. Additionally, the activity of this signaling axis in unperturbed cells has not been extensively investigated. We demonstrate that ATR promotes mTORC1 activity across various human cancer cells and both human and mouse normal cells under basal conditions. This effect is enhanced in human cancer cells (SKMEL28, RPMI-7951, HeLa) following knockdown of p16, a cell cycle inhibitor that we have previously found increases mTORC1 activity and here found increases ATR activity. Mechanistically, ATR promotes de novo cholesterol synthesis and mTORC1 activation through the phosphorylation and upregulation of lanosterol synthase (LSS), independently of both CHK1 and the TSC complex. Interestingly, this pathway is distinct from the regulation of mTORC1 by ATM and may be specific to cancer cells. Finally, ATR-mediated increased cholesterol correlates with enhanced localization of mTOR to lysosomes. Collectively, our findings demonstrate a novel connection linking ATR and mTORC1 signaling through the modulation of cholesterol metabolism.

**Keywords** Metabolism; Lanosterol Synthase; p16; Lysosome; Cholesterol
**Subject Category** Metabolism

## Introduction

The DNA damage response (DDR) plays an important role in maintaining genome stability (Jeggo et al, 2016). This is particularly apparent in cancer, where cells hijack DDR by upregulation of the key DDR proteins Ataxia Telangiectasia and Rad3-related protein (ATR) and/or Ataxia Telangiectasia Mutated (ATM) (Weber and Ryan, 2015). While activation of these pathways can induce cell cycle arrest to allow for DNA damage repair, cancers often have other cell cycle or DDR alterations that allow for continued proliferation even in the presence of activated ATR/ATM (Maréchal and Zou, 2013). Indeed, many cancers are often reliant on these proteins for continued proliferation, and ATR and ATM inhibitors are in clinical trials for a variety of malignancies (Priya et al, 2023). Additionally, recent work has demonstrated a critical role for ATR beyond the DDR in normal proliferation (Menolfi et al, 2023; Sugitani et al, 2022), suggesting that ATR activity under basal conditions plays an active role in cell signaling. However, pathways downstream of ATR beyond cell cycle control and DNA repair remain unclear.

DDR and metabolism are bidirectionally linked (Cucchi et al, 2021; Turgeon et al, 2018). We and others have shown that ATR and ATM have roles in metabolism (Aird et al, 2015; Chen et al, 2020; Dahl and Aird, 2017; Diehl et al, 2022; Huang et al, 2023), and metabolism can also influence DNA repair by regulating repair components and providing substrates necessary for DNA repair (Chatzidoukaki et al, 2020; Cucchi et al, 2021; Uboveja and Aird, 2024). It is therefore unsurprising that studies have previously identified a bidirectional relationship between both ATR and ATM to the master nutrient sensor and metabolic regulator mechanistic Target of Rapamycin Complex 1 (mTORC1) (Alexander et al, 2010; Danesh Pazhooh et al, 2021; Lamm et al, 2020; Ma et al, 2018), although the mechanisms linking these pathways have not yet been fully explored. We previously published that loss of the tumor suppressor p16 leads to mTORC1 hyperactivation in both normal

[1]Department of Pharmacology & Chemical Biology, University of Pittsburgh School of Medicine, Pittsburgh, PA, USA. [2]UPMC Hillman Cancer Center, University of Pittsburgh School of Medicine, Pittsburgh, PA, USA. [3]Center for Advanced Biotechnology and Medicine, Department of Pharmacology, and Rutgers Cancer Institute of New Jersey, Rutgers Robert Wood Johnson Medical School, Piscataway, NJ, USA. [4]Tsinghua University School of Medicine, Beijing, P.R. China. [5]Health Sciences Mass Spectrometry Core, University of Pittsburgh School of Medicine, Pittsburgh, PA, USA. [6]Department of Psychiatry, University of Pittsburgh School of Medicine, Pittsburgh, PA, USA. [7]Division of Malignant Hematology & Medical Oncology, Department of Medicine, University of Pittsburgh School of Medicine, Pittsburgh, PA, USA. [8]Department of Cardiovascular Sciences, Aging + Cardiovascular Discovery Center, Lewis Katz School of Medicine, Temple University, Philadelphia, PA, USA. [9]Department of Cell Biology, Center for Biologic Imaging (CBI), University of Pittsburgh, Pittsburgh, PA, USA. ✉E-mail: valvezan@cabm.rutgers.edu; katherine.aird@pitt.edu

and cancer cells (Buj et al, 2019). The mechanism underlying how p16 loss enhances mTORC1 activity in our model is not yet understood. We also found that p16 loss increases DNA damage (Tangudu et al, 2024). Whether the DDR and mTORC1 are linked downstream of p16 loss, or in other contexts, and the mechanism underlying this remains unclear.

mTORC1 is a master regulator of anabolic cell growth and proliferation that links upstream nutrient, growth factor, and energy signals to downstream metabolic pathways (Valvezan and Manning, 2019). mTORC1 activity is regulated through control of its spatial localization to the cytosolic surface of the lysosome. Sufficient levels of intracellular nutrients, including cholesterol, amino acids, and glucose, promote mTORC1 localization to the lysosome where it can be activated by the small GTPase Rheb (Castellano et al, 2017; Saxton and Sabatini, 2017; Shin et al, 2022). mTORC1 activation also requires inhibition of its upstream negative regulator, the TSC complex, which also resides at the lysosome. This often occurs downstream of growth factor signaling pathways, including PI3K/Akt, MEK/Erk, and others, which cause the TSC complex to dissociate from the lysosome, allowing mTORC1 activation (Ilagan and Manning, 2016; Menon et al, 2014). While ATR and ATM have been linked to mTORC1 activity (Alexander et al, 2010; Danesh Pazhooh et al, 2021; Lamm et al, 2020; Ma et al, 2018), whether and how these pathways affect mTORC1 lysosomal localization and activation is unknown.

Cholesterol is a key component of the plasma membrane and also plays important roles in signaling (Liu et al, 2023). The de novo cholesterol synthesis pathway is a complex, multi-step process in which acetyl-CoA is used to produce cholesterol through a series of intermediates including mevalonate, squalene, and lanosterol (Shi et al, 2022). Prior work has shown that cholesterol activates mTORC1 (Castellano et al, 2017; Davis et al, 2021; Lim et al, 2019; Shin et al, 2022), and cholesterol has also been linked to genome instability and the DDR (Liu et al, 2023). Whether the DDR and mTORC1 are linked through cholesterol has not been previously studied.

Here we observed that knockdown or inhibition of ATR decreases mTORC1 activity in multiple cell models to a greater extent than suppression of ATM, and that the ATR-mTORC1 signaling axis was especially apparent in p16 knockdown cells, which have mTORC1 hyperactivation (Buj et al, 2019). Using an unbiased proteomics approach cross-compared with publicly available datasets of potential upstream regulators of mTORC1, we identified the de novo cholesterol synthesis enzyme lanosterol synthase (LSS) as a mediator between ATR and mTORC1 activation in p16 knockdown cells. Moreover, ATR regulates LSS via phosphorylation in p16 knockdown cells. Indeed, p16 knockdown cells showed increased cholesterol abundance that is dependent on ATR. In multiple cancer cell models, suppression of mTORC1 activity by ATR inhibition or knockdown was rescued by exogenous cholesterol or lanosterol. Suppression of ATR also decreased mTOR localization at the lysosome, which was rescued by cholesterol. Together, our data demonstrate an unexpected link between ATR and mTORC1 via cholesterol synthesis in cancer cells.

# Results

## ATR and ATM suppression decrease mTORC1 activity in unperturbed cells independently of TSC2

Prior work has placed ATR and ATM upstream of mTORC1 under conditions of replication stress and/or DNA damage (Alexander et al, 2010; Danesh Pazhooh et al, 2021; Lamm et al, 2020; Ma et al, 2018). To explore whether ATR/ATM and mTORC1 are linked under unperturbed basal conditions, we inhibited or knocked down ATR and ATM in multiple cell models and assessed phosphorylation of the direct mTORC1 substrate S6K (pS6K) as a well-established readout of mTORC1 activity. Suppression of ATR or ATM, but not CHK1 or CHK2, decreased pS6K (Figs. 1A–C and EV1). This effect was also observed with siRNA-mediated knockdown of ATR, confirming the ATR inhibitor effects are not simply due to off-target effects on mTOR itself, which is a known effect of previous generation ATR inhibitors (Priya et al, 2023) (Fig. 1D). The TSC complex is a potent negative regulator of mTORC1 that integrates many upstream signals to control mTORC1 activity (Huang and Manning, 2008). Loss of the essential TSC complex component TSC2 did not affect ATR inhibitor-mediated reduction of pS6K, demonstrating the effect is independent of the TSC complex (Fig. 1E). CHK1 inhibition did not affect S6K phosphorylation regardless of TSC2 status (Fig. 1F). Surprisingly, mTORC1 activity was reduced by the dual CHK1/CHK2 inhibitor AZD7762; however, this appears to be an off-target effect as it was not reproduced with another dual CHK1/CHK2 inhibitor Prexasertib, or with combined CHK1 inhibitor plus CHK2 knockdown (Fig. EV1). Unlike ATR, the ATM inhibitor-mediated decrease in pS6K is dependent on the TSC complex (Fig. 1G). Together, these data demonstrate a link between the DDR and mTORC1 signaling in unperturbed cells and demonstrate that ATR-mediated mTORC1 activity is independent of the TSC complex and CHK1.

## ATR signaling downstream of p16 loss promotes mTORC1 activity

We found that suppressing ATR and ATM decreased mTORC1 activation in unperturbed cells (Fig. 1). Next, we aimed to test whether this pathway is intact in cells with combined high DDR and mTORC1 hyperactivation. Cells with knockdown of p16 have both increased DNA damage (Tangudu et al, 2024) and hyperactive mTORC1 signaling (Buj et al, 2019). Thus, we aimed to determine whether p16 knockdown cells increase mTORC1 activity via ATR and/or ATM using a previously validated shRNA against p16 (Buj et al, 2019). Consistent with the increase in DNA damage observed in p16 knockdown cells (Tangudu et al, 2024), we found that these cells have increased phosphorylation of ATR/ATM substrates pCHK1 and pCHK2 (Fig. EV2A). Additionally, using melanoma patient samples from TCGA, we found that homozygous deletion of CDKN2A, the gene encoding p16, correlated with increased pCHK1 along with other proteins involved in DNA damage response and repair (Fig. EV2B). Together, these data demonstrate that loss of p16 expression promotes ATR/ATM pathway activation, and this model is therefore useful towards understanding the mechanistic link between ATR/ATM and mTORC1. Knockdown or inhibition of ATR in p16 knockdown cells robustly decreased pS6K (Fig. 2A,B), while ATM knockdown only modestly decreased pS6K, and the ATM inhibitor KU60019 had no effect (Fig. 2C,D). This was not due to a lack of ATM inhibitor efficacy, as we noted robust downregulation of pCHK2 (Fig. 2D). Knockdown of p16 also strongly sensitized mTORC1 activity to ATR inhibition in HeLa cells (Fig. EV2C). Similar to unperturbed cells (Fig. 1A–C), knockdown of CHK1 in p16 knockdown cells did not robustly

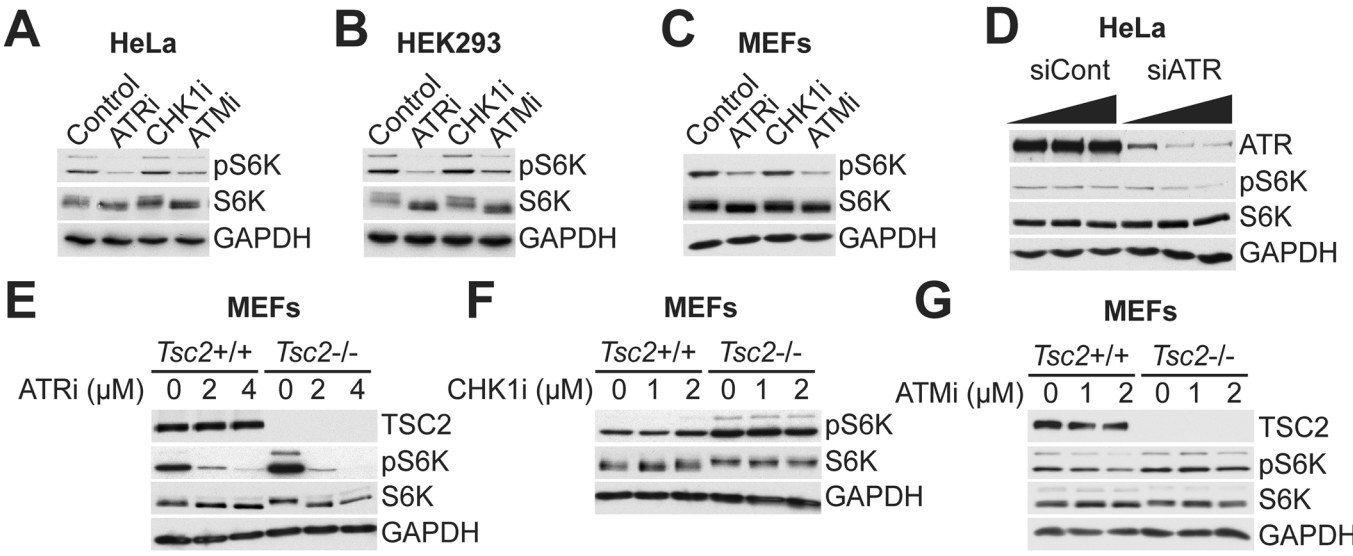

**Figure 1. ATR or ATM inhibition and knockdown suppress mTORC1 activity independent of TSC2.**

(A–C) HeLa cells (A), HEK293 cells (B), and MEFs (mouse embryonic fibroblasts) (C) were treated for 3 h with 2 µM AZD6738 ATR inhibitor (ATRi), 2 µM LY2603618 CHK1 inhibitor (CHK1i), or 5 µM KU55933 ATM inhibitor (ATMi), and the indicated proteins were assessed by western blotting. GAPDH was used as a loading control. (D) HeLa cells were transfected with siRNA against ATR or with a non-targeting siRNA as control, and the indicated proteins were assessed by western blotting. GAPDH was used as a loading control. (E–G) MEFs with TSC2 knockout were treated for 3 h with the indicated doses of AZD6738 ATR inhibitor (ATRi) (E), LY2603618 CHK1 inhibitor (CHK1i) (F), or KU55933 ATM inhibitor (ATMi) (G), and the indicated proteins were assessed by western blotting. GAPDH was used as a loading control. Data information: All western blots (A–G) are representative data from at least three independent experiments. Source data are available online for this figure.

decrease pS6K (Fig. EV2D), demonstrating that ATR promotion of mTORC1 activity is independent of this key downstream ATR substrate. Finally, we aimed to confirm the directionality of the effect as prior work has placed mTORC1 upstream of ATR (Danesh Pazhooh et al, 2021; Ma et al, 2018). Inhibition of mTORC1 signaling using Torin1 did not decrease pCHK1 in p16 knockdown cells (Fig. 2E). Together, these results demonstrate that ATR is upstream of mTORC1 and promotes its activation in p16 knockdown cells.

## ATR increases lanosterol synthase (LSS) and intracellular cholesterol

We next aimed to understand the mechanism that couples ATR to mTORC1 activity in p16 knockdown cells. We cross-compared proteomics data of p16 knockdown cells with knockdown of ATR to a publicly available dataset of potential mTORC1 regulators (Condon et al, 2021) (Fig. 3A). This resulted in 9 potential upstream mTORC1 regulators that were specific to p16 knockdown cells (Fig. 3B), which have high ATR-mediated mTORC1 activity (Fig. 2). The protein that was most significantly downregulated by ATR knockdown was lanosterol synthase (LSS) (Fig. 3B,C). LSS is an enzyme that forms the four-ring structure of cholesterol during the conversion of (S)-2,3-epoxysqualene to lanosterol in the de novo cholesterol synthesis pathway (Fig. 3D). We validated increased LSS in two different cell line models with p16 knockdown, which was subsequently reduced by ATR knockdown (Figs. 3E and EV3A). Consistent with ATR promoting LSS expression in p16 knockdown cells, we observed increased intracellular cholesterol levels using filipin staining that was subsequently decreased by ATR knockdown in both cell lines

(Figs. 3F,G and EV3B,C). We also found that inducing replication stress increased cholesterol levels, which was abrogated by ATR inhibition (Fig. EV3D,E), further supporting a role for ATR in regulating cholesterol.

We next aimed to determine the mechanism of increased LSS. This was not due to ATR effects on transcription, as knockdown of ATR had no effect on LSS mRNA expression (Fig. EV3F). ATR is a kinase, and LSS has five potential ATR/ATM consensus phosphorylation sites (S*/T*Q) (Fig. EV3G). Therefore, we aimed to determine whether LSS is a direct substrate of ATR. As there are no commercially available LSS phospho-specific antibodies, we immunoprecipitated LSS in p16 knockdown cells with or without ATR knockdown and assessed potential phosphorylation via immunoblotting using a Phospho-ATM/ATR Substrate antibody. We found that ATR knockdown reduces phosphorylation of LSS (Fig. 3H), suggesting that LSS is a direct ATR substrate. Together, these data indicate that ATR regulates LSS to increase cholesterol levels in p16 knockdown cancer cells.

## ATR promotes mTORC1 activity through LSS and cholesterol

We found that ATR increases LSS and cholesterol abundance (Fig. 3). Prior work found that cholesterol promotes mTORC1 activation (Castellano et al, 2017; Davis et al, 2021; Lim et al, 2019; Shin et al, 2022). Consistent with the ATR-LSS axis promoting mTORC1 activity via cholesterol in p16 knockdown cancer cells, knockdown or inhibition of either ATR or LSS decreased mTORC1 activity, and these effects were rescued by supplementation with either lanosterol or cholesterol (Fig. 4A–H). The decrease in mTORC1 activity upon ATR inhibition, but not ATM inhibition, in

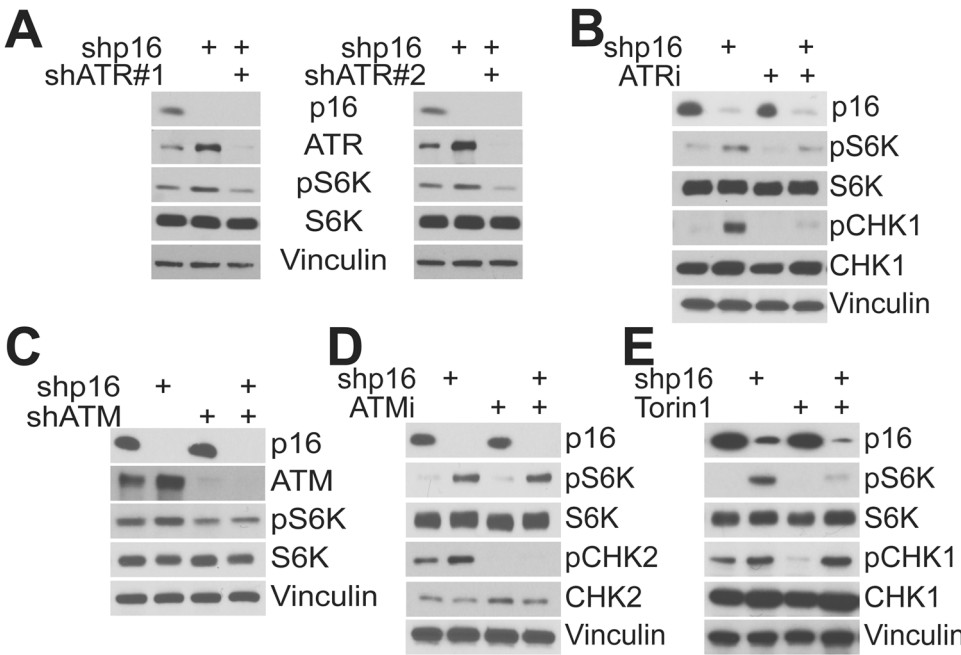

**Figure 2. ATR affects mTORC1 activity in p16 knockdown cells.**

(A–D) SKMEL28 cells were transduced with a lentivirus expressing a short hairpin RNA (shRNA) targeting GFP or p16 (shp16). (A) shp16 cells were transduced with a lentivirus expressing a shRNA targeting GFP or two independent shRNAs targeting ATR (shATR #1 and #2), and the indicated proteins were assessed by western blotting. Vinculin was used as a loading control. (B) Cells were treated for 30 min with 63 nM AZD6738 ATR inhibitor (ATRi), and the indicated proteins were assessed by western blotting. Vinculin was used as a loading control. (C) shp16 cells were transduced with a lentivirus expressing a shRNA targeting GFP or ATM (shATM), and the indicated proteins were assessed by western blotting. Vinculin was used as a loading control. (D) Cells were treated for 30 min with 313 nM KU60019 ATM inhibitor (ATMi), and the indicated proteins were assessed by western blotting. Vinculin was used as a loading control. (E) Cells were treated for 10 min with 250 nM of the mTORC1 inhibitor Torin1, and the indicated proteins were assessed by western blotting. Vinculin was used as a loading control. Data information: All western blots (A–G) are representative data from at least three independent experiments. Source data are available online for this figure.

another cancer cell line (HeLa) was also rescued by supplementation with cholesterol (Fig. EV4A,B). Interestingly, cholesterol supplementation did not rescue pS6K in normal cells upon ATR inhibition (Fig. EV4C,D), indicating that cancer cells can regulate mTORC1 via ATR and cholesterol, whereas normal cells may have an alternative pathway. Together, our data demonstrate that ATR upregulates LSS, which promotes mTORC1 activity through cholesterol in cancer cells.

## ATR promotes mTOR localization to lysosomes via cholesterol

mTOR localization to the lysosome in response to nutrients facilitates its activation (Castellano et al, 2017; Kim et al, 2008; Sancak et al, 2008). Thus, we aimed to determine whether ATR promotes mTOR lysosomal localization through its effects on cholesterol. Indeed, knockdown or inhibition of ATR decreased mTOR localization to the lysosome, and this effect was rescued by exogenous cholesterol supplementation (Figs. 5A and EV5A). Consistent with ATR promoting mTORC1 activation independent of the TSC complex (Fig. 1E), ATR knockdown or inhibition did not increase TSC2 lysosomal localization (Figs. 5B and EV5B). Surprisingly, ATR knockdown slightly reduced TSC2 lysosomal localization in p16 knockdown SKMEL28 cells, although this did not occur with ATR inhibition in HeLa cells and was not affected by cholesterol (Figs. 5B and EV5B). Taken together with results in

Figs. 3 and 4, these data put forth a model in which ATR promotes LSS expression and increased intracellular cholesterol levels, which activate mTORC1 by promoting its lysosomal localization (Fig. 6).

## Discussion

Prior work found that ATR is upstream of mTORC1 under replication stress (Lamm et al, 2020). However, whether ATR influences mTORC1 in unperturbed cells is unknown. Moreover, mTORC1 is a central nutrient sensor, and the metabolic mechanism linking ATR to mTORC1 has not been described. Here we found that ATR promotes mTORC1 activation both in unperturbed cells and those with p16 knockdown. This is independent of both CHK1 and the TSC complex, and in cancer cells occurs through LSS regulation by ATR to increase cholesterol levels. We validated these effects using both siRNA/shRNA-mediated knockdown of ATR, as well as with the ATR inhibitor AZD6738. Both ATR and mTOR belong to the PI3K-like kinase (PIKK) family, and previous generation ATR inhibitors have known off-target effects on mTOR, underscoring the importance of ATR knockdown in this study. An off-target effect of AZD6738 on mTOR can also be ruled out by the rescue with lanosterol/cholesterol in cancer cells (Figs. 4 and EV4), and by the sensitization upon p16 knockdown (Fig. EV4). Thus, these results

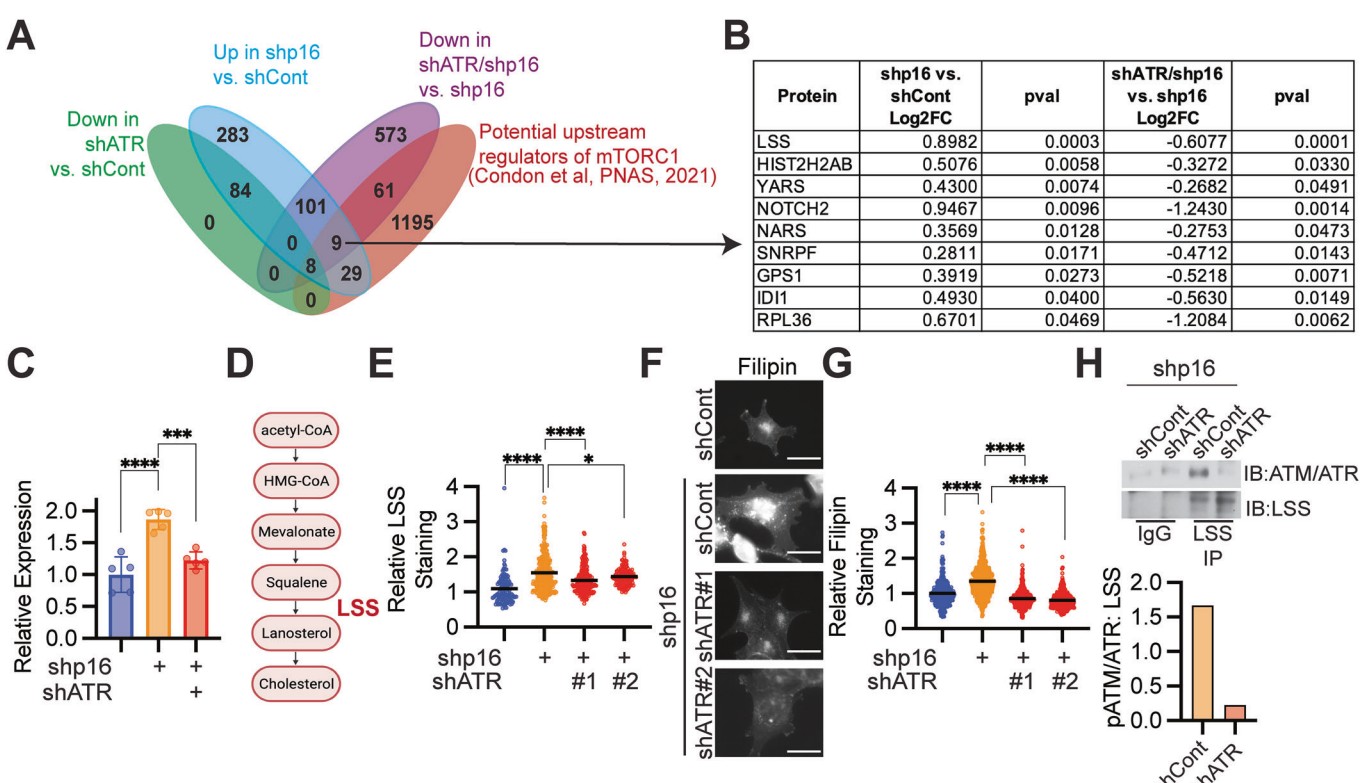

**Figure 3. ATR decreases lanosterol synthase expression and phosphorylation, corresponding to decreased cholesterol abundance.**

(A–G) SKMEL28 cells were transduced with lentivirus expressing shRNA targeting GFP (shCont) or p16 (shp16) with or without lentivirus expressing shRNA targeting ATR (shATR #1 and #2). (A) Venn diagram comparing proteomics data with publicly available dataset of mTORC1 regulators. (B) Nine hits were identified in the cross-comparison from (A). (C) Relative LSS protein abundance in the indicated groups was determined by mass spectrometry (n = 5). The graph represents mean ± SD. ***p = 0.0006, ****p < 0.0001. (D) Simplified pathway of de novo cholesterol synthesis and LSS. Created in BioRender. Aird, K. (2025) https://BioRender.com/t5fzdhy. (E) LSS expression was assessed by immunofluorescence staining and quantified. The graph represents individual normalized values and the mean. *p = 0.0376, ****p < 0.0001. (F, G) Cholesterol abundance was assessed by filipin staining (F) and quantified (G). Scale bar = 20 μm. The graph represents individual normalized values and the mean. ****p < 0.0001. (H) shp16 SKMEL28 cells were transduced with lentivirus expressing shRNA targeting ATR (shATR). Protein lysates were immunoprecipitated with anti-LSS antibody and ATM/ATR substrate, and LSS were assessed by western blotting (upper panel). Integrated density analysis of ATM/ATR substrate normalized to LSS (lower panel). Representative data from one of three independent experiments is shown. Data information: Proteomics in (B, C) was repeated one time with five technical replicates. Data in (E–H) are representative data from at least three independent experiments. Statistical analysis in (C, E, G) was performed using one-way ANOVA with Šídák's multiple comparisons test. Source data are available online for this figure.

firmly establish a link between ATR, cholesterol signaling, and mTORC1 activity in cancer cells.

There is a bidirectional cross-talk between mTORC1 and DNA damage. Interestingly, most studies have shown that mTORC1 influences the DNA damage response (Danesh Pazhooh et al, 2021; Ma et al, 2018), with only a handful demonstrating that ATR or ATM are upstream of mTORC1 activation (Alexander et al, 2010; Lamm et al, 2020). One prior report found that replication stress, and therefore ATR activity, promotes nuclear mTORC1 activation (Lamm et al, 2020), although no direct mechanism was identified and whether mTORC1 can be activated outside of the lysosome remains an open question. Here we found that ATR promotes mTORC1 activation (Figs. 1 and 2), through increasing cholesterol levels and mTORC1 localization to the lysosome (Figs. 3–6). DNA damage has also been shown to induce or rewire lipid metabolism (Carroll et al, 2015; Hammerquist et al, 2021; Hamsanathan et al, 2022; Xiao et al, 2020), although to our knowledge, no prior studies have examined cholesterol in this context. Here we found that inducing replication stress in parental cancer cells or shp16 cells

with high endogenous DNA damage (Tangudu et al, 2024) increased cholesterol levels in an ATR-dependent manner (Figs. 3F,G and EV3B–D). Mechanistically, we found that ATR increases expression of LSS, a critical enzyme in de novo cholesterol synthesis (Fig. 3). We found that LSS is a potential direct substrate of ATR (Fig. 3H). However, which serine or threonine sites are phosphorylated and whether this phosphorylation event affects downstream signaling remains to be investigated in future studies. Phosphorylation may also affect LSS activity or the translocation of the protein. Interestingly, prior high throughput proteomics studies found LSS in or near the nucleus (MacDonald et al, 2024; Wang et al, 2017), while it is canonically thought to be at the endoplasmic reticulum (Romano et al, 2018). Moreover, ATR was long thought of as a nuclear protein but is has also recently been found in the cytoplasm (Biswas et al, 2022; Hilton et al, 2016; Postigo et al, 2017). Therefore, we speculate that the interaction between ATR and LSS is likely not at the lysosome but in or near the nucleus. We observed that ATR regulates mTORC1 via cholesterol in cancer cells but not normal cells (Figs. 4 and EV4). It is interesting to

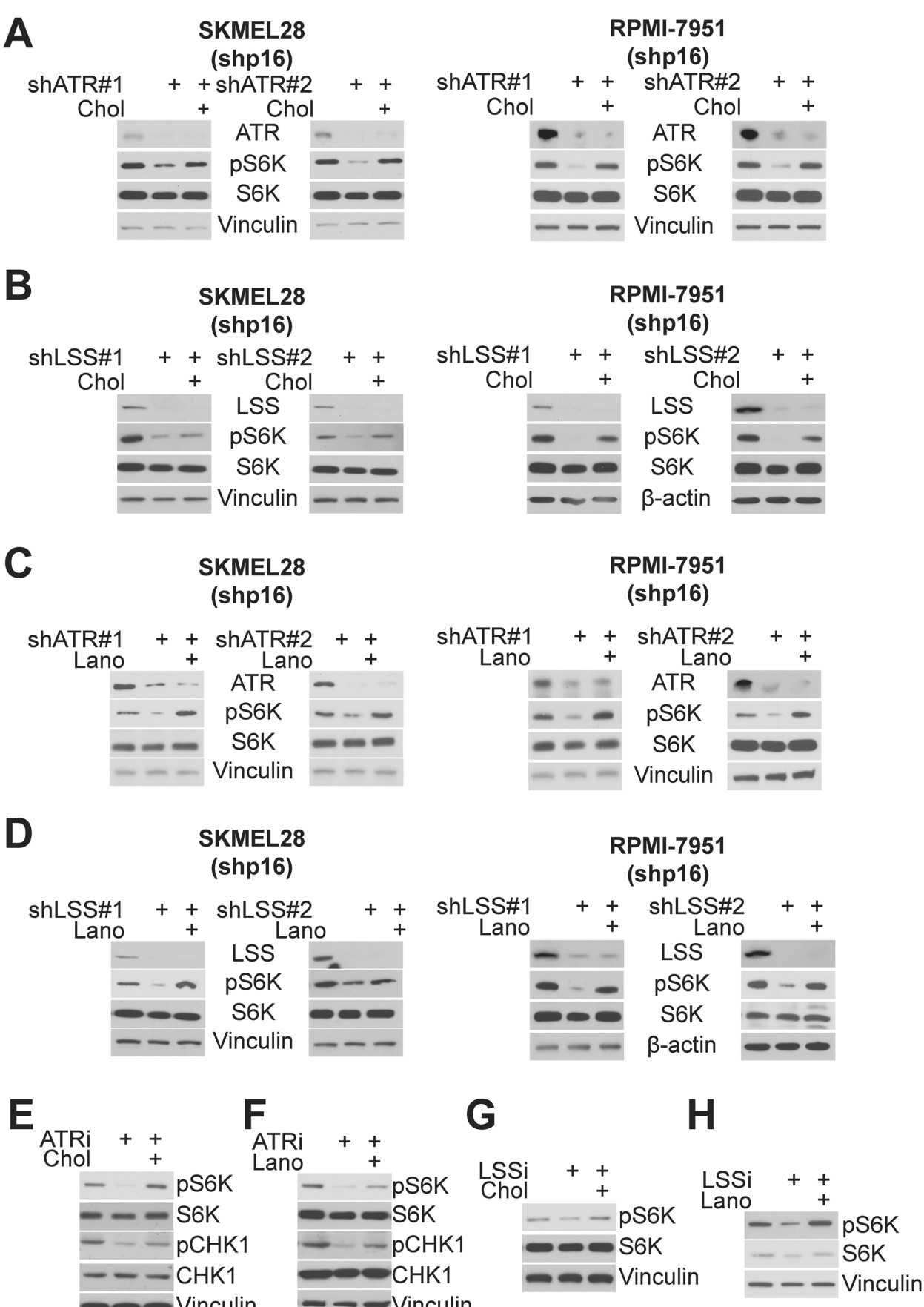

**Figure 4. ATR increases mTORC1 activity via cholesterol and lanosterol.**

(A–D) The indicated shp16 cells were transduced with lentivirus expressing shRNA targeting ATR (shATR #1 and #2) (A, C) or targeting LSS (shLSS #1 and #2) (B, D). Cells were supplemented with 50 μM cholesterol or lanosterol, where indicated. The indicated proteins were assessed by western blotting. (E, F) shp16 SKMEL28 cells were treated with the ATRi and supplemented with 50 μM cholesterol (E) or lanosterol (F). (G, H) shp16 SKMEL28 cells were treated with the LSSi and supplemented with 50 μM cholesterol (G) or lanosterol (H). Data information: Western blot data in (A–D) are representative data from two to three independent experiments in each cell line. Western blot data in (E–H) are representative data from at least three independent experiments. Source data are available online for this figure.

speculate that ATR interaction with LSS in different cellular compartments may be cell-type-dependent. Additionally, this observation may be due to lipid metabolic derangement, elevated cholesterol uptake, or high basal replication stress of cancer cells, which will be investigated in future studies.

We found that ATR suppresses mTORC1 in part via reducing cholesterol levels, as cholesterol supplementation rescued the effect of ATR knockdown or inhibition on mTORC1 (Fig. 4). We demonstrated that this effect is through changes in mTOR localization at the lysosome, and independent of the TSC complex (Fig. 5). Prior work has elegantly shown how cholesterol promotes mTORC1 activity at the lysosome via LYCHOS (Shin et al, 2022). Whether LYCHOS mediates the changes in mTORC1 activity downstream of ATR remains to be tested, although it is likely given that LYCHOS is the well-established cholesterol sensor at the lysosome for the mTORC1 network, and we observed increased lysosome-associated mTOR in the context of high cholesterol (Fig. 5). It is also interesting to speculate whether increased cholesterol has other signaling effects beyond increasing mTORC1 activity. Cholesterol is a critical component of the plasma membrane but can also lead to increased steroid hormones, vitamin D, and bile acids (Centonze et al, 2022; Liu et al, 2023). We and others have found that p16 knockdown cells proliferate faster than controls (Fung et al, 2013; Tangudu et al, 2024), and thus likely need increased membrane synthesis to support more frequent cell division. Additionally, both steroids and bile acids promote pro-tumorigenic signaling pathways (Centonze et al, 2022; Liu et al, 2023; Tangudu et al, 2024). Bile acids also induce ROS (Orozco-Aguilar et al, 2021), which may induce and/or reinforce the DNA damage observed in p16 knockdown cells (Tangudu et al, 2024). Together, this likely indicates that cholesterol metabolism may have pleotropic effects on these cells and point to a multifaceted role for the ATR-LSS axis in cancer biology.

ATR inhibitors are currently in clinical trials (Priya et al, 2023). Our study suggests that there may be potential metabolic effects of these drugs related to mTORC1 signaling and cholesterol synthesis. Importantly, ATR's effects on mTORC1 via cholesterol appear to be specific to cancer cells, which may indicate that these inhibitors will more specifically target cholesterol synthesis in cancer cells and not normal tissue. Indeed, that would be value added since cholesterol has pro-tumor effects on both cancer cells and the tumor microenvironment (Guo et al, 2025; Liu et al, 2023). Additionally, statins and LSS deficiency have been shown to potentiate immunotherapy in preclinical models (Cantini et al, 2021; Gao et al, 2024; Kansal et al, 2023; Liao et al, 2025; Mao et al, 2022; Takada et al, 2022), phenocopying ATR inhibitors (Hardaker et al, 2024; Ngoi et al, 2022; Sheng et al, 2020; Taniguchi et al, 2024). Future studies to determine whether ATR's regulation of cholesterol and mTORC1 affects immunotherapy response are therefore of great interest. We also believe there is a need for future

work to determine whether the effects of ATR inhibitors to suppress mTORC1 in normal cells, especially immune cells, is counteractive to its effects on the cancer cells themselves. Optimizing dosing schedules for ATR inhibitors to minimize prolonged suppression of cholesterol synthesis and mTORC1 signaling in normal cells may therefore be crucial.

In summary, we report a novel link between ATR and mTORC1 via LSS and cholesterol, which has many important implications. In cancer, ATR inhibitors are currently under clinical development (Priya et al, 2023), and our results demonstrate that these inhibitors will have pleotropic effects outside of the DNA damage response that may influence therapeutic efficacy. It also raises the interesting possibility that previously reported effects of ATR on metabolism could be mediated through its effects on mTORC1. Moreover, since ATR signals to mTORC1 in normal cells (Fig. 1), although not through cholesterol (Fig. EV4), our data suggest that there may be side effects of ATR inhibitors on mTORC1 signaling that should be considered in clinical application of these therapeutics. Additionally, the DDR and mTORC1 are both deregulated during aging (Johnson et al, 2013; Schumacher et al, 2021). Thus, it is interesting to speculate whether cholesterol also connects these signaling pathways during aging or other pathologies associated with increased DNA damage. Finally, recent work has found that ATR plays an important role in normally proliferating cells, such as T cells (Menolfi et al, 2023; Sugitani et al, 2022). Therefore, this pathway may be of broad relevance to normal physiology.

# Methods

**Reagents and tools table**

| Reagent/resource | Reference or source | Identifier or catalog number |
| --- | --- | --- |
| **Experimental models** | | |
| SKMEL28 | ATCC | Cat # HTB-72 |
| RPMI-7951 | ATCC | Cat # HTB-66 |
| HeLa | ATCC | Cat # CCL-2 |
| HEK293-FT | ATCC | Cat # CRL-3249 |
| HEK293 | ATCC | Cat # CRL-1573 |
| MEFs | Provided by DJ Kwiatkowski | Zhang et al, 2003 |
| **Recombinant DNA** | | |
| pLKO.1 puro | Addgene | Cat # 8453 |
| pLKO.1 GFP | Addgene | Cat # 30323 |
| psPAX2 | Addgene | Cat # 12260 |
| pMD2.G | Addgene | Cat # 12259 |

| Reagent/resource | Reference or source | Identifier or catalog number |
|---|---|---|
| **Antibodies** | | |
| anti-CDKN2A/p16INK4a | Abcam | Cat # ab108349 |
| anti-Phospho-Chk1 (Ser345) | Cell Signaling Technology | Cat # 2348 |
| anti-Phospho-Chk1 (Ser296) | Cell Signaling Technology | Cat # 2349 |
| anti-Chk1 | Cell Signaling Technology | Cat # 2360 |
| anti- Phospho-p70 S6 Kinase (T389) | Cell Signaling Technology | Cat # 9234 |
| anti- p70 S6 Kinase | Cell Signaling Technology | Cat # 2708 |
| anti-Phospho-Chk2 (T68) | Cell Signaling Technology | Cat # 2197 |
| anti-Chk2 | Cell Signaling Technology | Cat # 2662 |
| anti-LSS | Thermo Fisher Scientific | Cat # 18693-1-AP |
| anti-LSS | Sigma Aldrich | Cat # HPA032060 |
| anti-β-actin | Sigma Aldrich | Cat # A1978 |
| anti-vinculin | Sigma Aldrich | Cat # V9131 |
| anti-ATR | Bethyl laboratories | Cat # A300-138A |
| anti-Rabbit IgG | Cell Signaling Technology | Cat # 2729 |
| anti-mouse IgG, HRP-linked | Cell Signaling Technology | Cat # 7076 |
| anti-rabbit IgG, HRP-linked | Cell Signaling Technology | Cat # 7074 |
| anti-TSC2 | Cell Signaling Technology | Cat # 4308 |
| anti-mTOR | Cell Signaling Technology | Cat # 2938 |
| anti-LAMP2 | Santa Cruz | Cat # sc18822 |
| anti-Mouse IgG (H + L), AF488 | Thermo Fisher Scientific | Cat # A-11001 |
| anti-Rabbit IgG (H + L), Cy3 | Jackson Immuno | Cat # 11-165-144 |
| **Oligonucleotides and other sequence-based reagents** | | |
| CDKN2A shRNA | Horizon discovery | Cat # TRCN0000010482 |
| ATR shRNA #1 | Open Biosystems | Cat # TCRN0000039615 |
| ATR shRNA #2 | Open Biosystems | Cat # TCRN0000039616 |
| ATM shRNA | Open Biosystems | Cat # TCRN0000038658 |
| CHK1 shRNA #1 | Sigma Aldrich | Cat # TRCN0000000499 |
| CHK1 shRNA #2 | Sigma Aldrich | Cat # TRCN0000000502 |
| LSS shRNA #1 | Sigma Aldrich | Cat # TRCN0000045481 |
| LSS shRNA #2 | Sigma Aldrich | Cat # TRCN0000045482 |
| CHK2 siRNA | Horizon Discovery/ Dharmacon | Cat # L-003256-00-0005 |
| **Chemicals, enzymes and other reagents** | | |
| DMEM | Fisher Scientific | Cat # MT10013CV |
| Fetal bovine serum | BioWest | Cat # S1620 |

| Reagent/resource | Reference or source | Identifier or catalog number |
|---|---|---|
| Lipid-depleted fetal bovine serum | Omega Scientific | Cat # FB-50 |
| Methyl-β-cyclodextrin | Sigma Aldrich | Cat # C4555 |
| Cholesterol | Sigma Aldrich | Cat # C3045 |
| Lanosterol | Sigma Aldrich | Cat # L5768 |
| Lipofectamine™ 2000 | Invitrogen | Cat # 11668019 |
| Opti-MEM | Gibco | Cat # 31985070 |
| AZD6738 | Selleckchem | Cat # S7693 |
| Ro 48-8071 | Cayman Chemical | Cat # 10006415 |
| KU55933 | Selleckchem | Cat # S1092 |
| KU60019 | APEX bio | Cat # A8336 |
| LY2603618 | Selleckchem | Cat # S2626 |
| AZD7762 | Selleckchem | Cat # S1532 |
| Prexasertib/LY2606368 | Medchem express | Cat # HY-18174 |
| Cycloheximide | Sigma Aldrich | Cat # C7698 |
| p-phenylenediamine | EMD chemicals | Cat # PX0730 |
| Intercept blocking buffer | Licor | Cat # 927-70001 |
| Fluoromount G | SothernBiotech | Cat # 0100-01 |
| Trizol | Ambion | Cat # 15596018 |
| EDTA-free protease inhibitor cocktail | Sigma Aldrich | Cat # 11836170001 |
| Phosphatase inhibitor cocktail 2 | Sigma Aldrich | Cat # P5726 |
| Phosphatase inhibitor cocktail 3 | Sigma Aldrich | Cat # P0044 |
| PureProteome™ Protein G Magnetic Bead | Sigma Aldrich | Cat # LSKMAGG10 |
| SuperSignal West Pico PLUS Chemiluminescent Substrate | Thermo Fisher Scientific | Cat # 34580 |
| Hydroxyurea | Sigma Aldrich | Cat # H8627 |
| Filipin III | Cayman Chemical | Cat # 70440 |
| Bio-Rad Protein Quantification reagent | Bio-Rad | Cat # 5000006 |
| **Software** | | |
| GraphPad Prism 10.0 | https://www.graphpad.com, Dotmatics | |
| Cellpose v. 3.0.7 | https://www.cellpose.org, Stringer et al, 2021 | |
| MaxQuant software suite (version 2.1.3) | https://www.maxquant.org, Cox and Mann, 2008 | |
| Andromeda protein identification search engine | Cox et al, 2011 | |
| cBioportal | https://www.cbioportal.org | |
| ZenBlue 3.3 analysis suite | Zeiss | |
| **Other** | | |
| Next-generation sequenicng | Novogene | |

| Reagent/resource | Reference or source | Identifier or catalog number |
|---|---|---|
| BCA protein quantification kit | Thermo Fisher Scientific | Cat # 23225 |
| BioAnalyzer RNA 6000 Nano Kit | Agilent Technologies | Cat # 5067-1511 |

## Cell lines

SKMEL28, RPMI-7951, and HeLa cells were purchased from ATCC. SKMEL28 cells were cultured in DMEM (Fisher Scientific, cat# MT10013CV) supplemented with 5% fetal bovine serum (BioWest, cat# S1620). RPMI-7951 cells were cultured in MEM (Fisher Scientific cat#MT10009CV) supplemented with 5% fetal bovine serum (BioWest, cat# S1620). HEK293 and HeLa cells were cultured in DMEM (Fisher Scientific cat# MT10013CV) supplemented with 5% fetal bovine serum (BioWest, cat# S1620). MEFs were cultured in DMEM (Fisher Scientific cat# MT10013CV) supplemented with 5% fetal bovine serum (BioWest, cat# S1620). Cells were supplemented with 1% Penicillin/Streptomycin (Fisher Scientific, cat#15-140-122). All cell lines were routinely tested for mycoplasma as described in (Uphoff and Drexler, 2005).

## Lentiviral packaging and infection

Lentiviral vectors were packaged using the Lipofectamine™ 2000 Transfection Reagent (Invitrogen, cat# 11668019). Briefly, the target lentiviral vector was mixed with 7.5 μg of psPAX2 (Addgene, cat# 12260) and 5 μg of pMD2.G (Addgene, cat# 12259) in 1.5 mL of Opti-MEM™ media (Gibco, cat# 31985070). The Lipofectamine-lentiviral complexes were then added dropwise onto HEK293-FT cells (ATCC, cat# CRL-3249). Supernatant containing viral particles was collected 72 h after transfection. Cells were transduced with corresponding vectors for 16 h and selected for 3 days with 1–3 μg/ml puromycin. The plasmids are the following: pLKO.1-shp16 (TRCN0000010482); pLKO.1-shGFP control (Addgene, cat# 30323); pLKO.1-shATR (TCRN0000039615, TCRN0000039616); pLKO.1-shATM (TCRN0000038658); pLKO.1-shChk1 (TRCN0000000499, TRCN0000000502); pLKO.1-LSS (TRCN0000045481, TRCN0000045482).

## Western blotting

Cells lysates were collected in 1X sample buffer (2% SDS, 10% glycerol, 0.01% bromophenol blue, 62.5 mM Tris, pH 6.8, 0.1 M DTT) or lysis buffer containing 20 mM Tris pH 7.5, 140 mM NaCl, 1 mM EDTA, 10% glycerol, 1% Triton X-100, 50 mM NaF, 1 mM DTT, with protease inhibitor cocktail (Sigma Aldrich, cat# P8340), phosphatase inhibitor cocktail #2 (Sigma Aldrich, cat# P5726), and #3 (Sigma Aldrich, cat# P0044) used at 1:100 each, and then boiled to 95 °C for 5–10 min. Cell lysates were sonicated for 10–15 s. Protein concentration was determined using the Bio-Rad protein quantification reagent (Bio-Rad, cat# 5000006). An equal amount of total protein was resolved using SDS-PAGE gels and transferred to nitrocellulose membranes (Fisher Scientific, cat# 10600001) at 110 mA for 2 h or overnight at 4 °C. Membranes were blocked with 5% nonfat milk or 4% BSA in TBS containing 0.1% Tween-20 (TBS-T) for 1 h at room temperature. Membranes were incubated overnight at 4 °C in primary antibodies in 4% BSA/TBS + 0.025% sodium azide. Membranes were washed four times in TBS-T for 5 min at room temperature after which they were incubated with HRP-conjugated secondary antibodies (Cell Signaling Technology) for 1 h at room temperature. After washing four times in TBS-T for 5 min at room temperature, proteins were visualized on film after incubation with SuperSignal West Pico PLUS Chemiluminescent Substrate (Thermo Fisher Scientific, cat# 34580). Primary antibodies: Rabbit anti-p16 INK4A (Abcam, cat# ab108349; 1:1000); Phospho-Chk1 (Ser345) (Cell Signaling Technology, Cat# 2348, 1:1000); Phospho-Chk1 (Ser296) (Cell Signaling Technology, Cat# 2349, 1:1000); Mouse anti-Chk1 (Cell Signaling Technology, Cat# 2360, 1:1000); Rabbit anti-Phospho-Chk2 (Thr68) (Cell Signaling Technology, Cat 2197, 1:1000); Rabbit anti-Chk2 (Cell Signaling Technology, Cat# 2662, 1:1000); Rabbit anti-LSS (Sigma Aldrich, cat# HPA032060; 1:1000); Rabbit anti-LSS (Thermo Fisher Scientific, cat# 18693-1-AP; 1:1000); mouse anti-β-actin (Sigma Aldrich, cat# A1978; 1:10,000); Mouse anti-vinculin (Sigma Aldrich, cat# V9131; 1:10,000); Rabbit anti-ATR (Bethyl, Cat # A300-138A, 1:1000); Rabbit anti-phopsho-p70 S6 Kinase (T389) (Cell Signaling Technology, Cat # 9234, 1:1000); Rabbit anti-p70 S6 Kinase (Cell Signaling Technology, Cat # 2708, 1:1000); Secondary antibodies: Anti-mouse IgG, HRP-linked (Cell Signaling Technology, cat# 7076; 1:10,000 and 1:5000), Anti-Rabbit IgG, HRP-linked (Cell Signaling Technology, cat# 7074; 1:5000).

## Analysis of the Cancer Genome Atlas patient data

RPPA data were extracted from the Cancer Genome Atlas (TCGA) skin cutaneous melanoma firehose legacy using cBioportal. Patients were divided based on homozygous deletion (or not) of CDKN2A.

## Proteomics

SKMEL28 cells were homogenized in 50 mM TEAB, 5% SDS. Total protein was measured by microBCA (Pierce). About 500 μg total protein was digested on S-Trap Midi's (Protifi) per the manufacturer's protocol and desalted on Peptide Desalting Spin Columns (Pierce). Phosphopeptides were enriched on an AssayMAP Bravo (Agilent) with an Fe3+ column. LC-TIMS-MS/MS analysis was carried out using a nanoElute UHPLC system (Bruker Daltonics, Bremen, Germany) coupled to the timsTOF Pro mass spectrometer (Bruker Daltonics), using a CaptiveSpray nanoelectrospray (Bruker Daltonics). Roughly 100 ng of peptide digest or phosphopeptide enrichment was loaded on a capillary C18 column (25 cm length, 75-μm-inner diameter, 1.6-μm particle size, 120 Å pore size; IonOpticks, Fitzroy, VIC, AUS). Peptides were separated at 55 °C using a 60 min gradient at a flow rate of 300 nL/min (mobile phase A (MPA): 0.1% FA; mobile phase B (MPB): 0.1% FA in acetonitrile). A linear gradient of 2–35% MPB was applied for 60 min, followed by a 5 min wash at 95% MFB before equilibrating the column at 2% MFB for 6 min. The timsTOF Pro was operated in PASEF mode, collecting full scan mass spectra from 100 and 1700 $m/z$. Ion mobility resolution was set to 0.60–1.60 V·s/cm over a ramp time of 100 ms. Data-dependent acquisition was performed using 10 PASEF MS/MS scans per 1.1 s

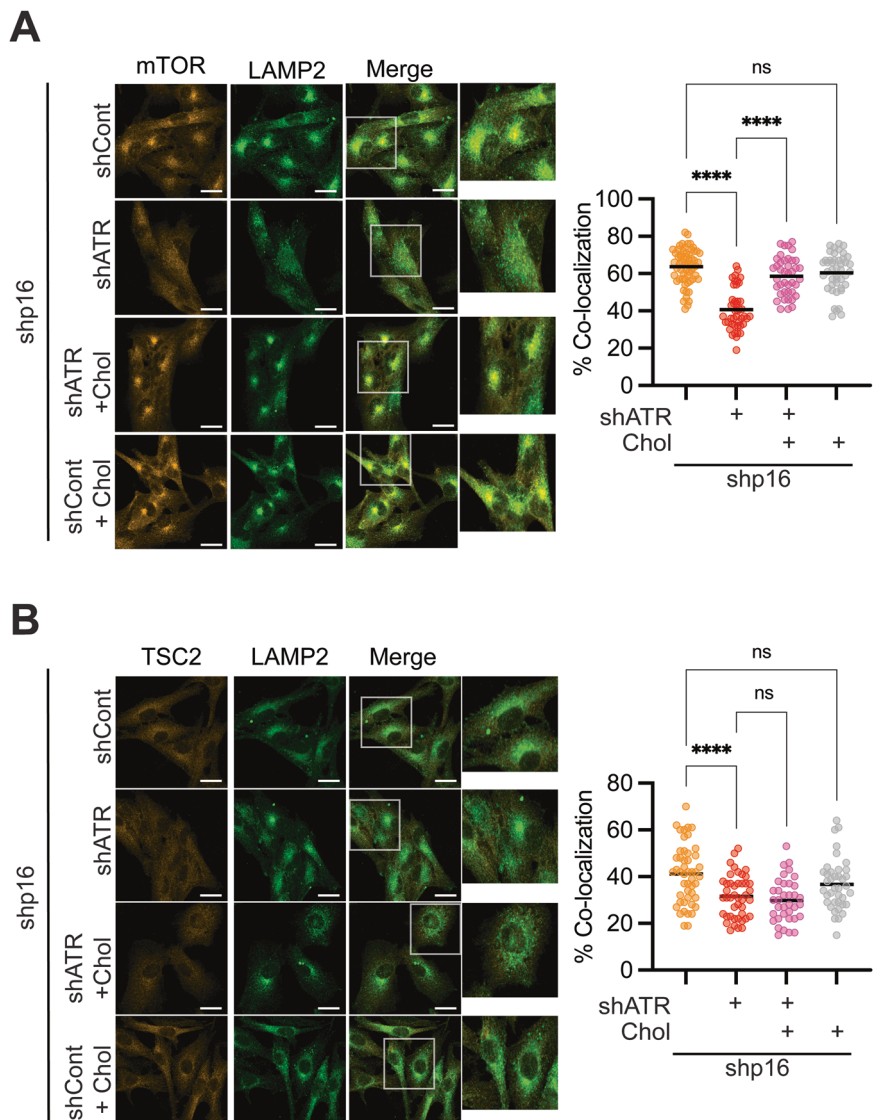

**Figure 5.  ATR promotes mTOR localization to lysosomes via cholesterol.**

(A, B) shp16 SKMEL28 cells were transduced with lentivirus expressing shRNA targeting ATR (shATR). Where indicated, cells were supplemented with 50 µM cholesterol. (A) Representative images of mTOR and LAMP2 immunofluorescence staining (left), which is quantified on the right. The graph represents individual values and the mean. Scale bar = 20 µm. ****$p$ < 0.0001, ns = not significant. (B) Representative images of TSC2 and LAMP2 immunofluorescence staining (left), which is quantified on the right. The graph represents individual values and the mean. Scale bar = 20 µm. ****$p$ < 0.0001, ns not significant. Data information: Representative data from one of three independent experiments is shown. Graphs represent individual values and the mean. Statistical analysis in (A, B) was performed using One-way ANOVA with Šídák's multiple comparisons test. ns not significant, ****$p$ < 0.0001 Source data are available online for this figure.

cycle. The active exclusion time window was set to 0.4 min, and the intensity threshold for MS/MS fragmentation was set to 2.5e4 while low $m/z$ and singly charged ions were excluded from PASEF precursor selection. MS/MS spectra were acquired via ramped collision energy as a function of ion mobility.

The nLC-MS/MS data were analyzed with the MaxQuant software suite (version 2.1.3) (Cox and Mann, 2008). The Andromeda protein identification search engine (Cox et al, 2011) and a SwissProt human protein database (downloaded on November 11, 2022, with 20,403 entries) were utilized with default settings for Orbitrap instruments. The parameters used included a precursor mass tolerance of 20 ppm for the first search and 4.5 ppm

for the main search, a product ion mass tolerance of 0.2 Da, and a minimum peptide length of five amino acids. Trypsin was set as the proteolytic enzyme with a maximum of two missed cleavages allowed. The enzyme specifically cleaves peptide bonds C-terminal of arginine and lysine if they are not followed by proline. Carbamidomethylation of cysteine was set as a fixed modification. Oxidation of methionine, deamination of both asparagine and glutamine, and acetylation of the protein N-terminus were set as variable modifications. Phosphorylation of serine, threonine, and tyrosine was set as a variable modification for phosphopeptides. A 1% false discovery rate (FDR) was used to filter the peptide identification results. The integrated feature intensities provide a

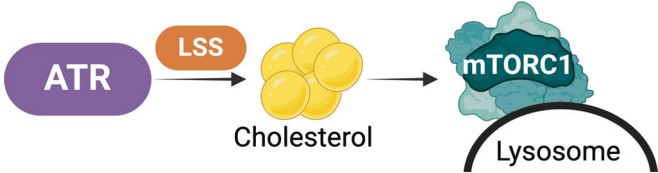

**Figure 6. Schematic of the proposed model linking ATR to mTORC1 signaling via de novo cholesterol synthesis.**

ATR increases LSS to promote cholesterol synthesis, which drives mTOR to the lysosome and increases its activity. Created in BioRender. Aird, K. (2025) https://BioRender.com/k7rrz72.

relative measure of abundance for each feature at the peptide level and are used in all subsequent analyses. Proteomics data can be found in the Source Data.

## Cholesterol and lanosterol treatment

Cells were seeded at an equal density on plates or coverslips. Post-lentiviral infection, cells were selected with puromycin for 24 h. Cells were washed once with DMEM and starved in 0.5% lipid-depleted FBS/0.75%-MβCD (Omega Scientific, cat# FB-50/Sigma Aldrich, Cat# C4555) contained DMEM for 2 h. Cells were washed once with DMEM and supplemented with 50 µM Cholesterol (Sigma Aldrich, Cat# C3045) or lanosterol (Sigma Aldrich, Cat# L5768)/0.1%-MβCD/0.5% lipid-depleted FBS for 2 h.

## Immunofluorescence

Cells were seeded at an equal density on coverslips and fixed with 4% paraformaldehyde. Cells were washed four times with PBS and permeabilized with 0.2% Triton X-100 in PBS for 5 min. For LSS staining, cells were blocked for 5 min with 3% BSA/PBS, followed by incubation of the corresponding Rabbit anti-LSS primary antibody (Sigma Aldrich, cat# HPA032060; 1:200) in 3% BSA/PBS for 1 h at room temperature. Cells were washed three times with 1% Triton X-100 in PBS and incubated with secondary antibody (Cy3 Goat Anti-Rabbit (Jackson Immuno, cat# 111-165-003, 1:5000), and Cy3-AffiniPure Donkey Anti-Mouse (Jackson Immuno, cat# 715-165-150, 1:5000)) in 3% BSA/PBS for 1 h at room temperature. Cells were then incubated with 0.15 µg/ml DAPI for 1 min, washed three times with PBS, mounted with fluorescence mounting medium (9 ml of glycerol [Fisher Scientific, cat# BP229-1], 1 ml of 1× PBS, and 10 mg of p-phenylenediamine [EMD Chemicals, cat# PX0730]; pH was adjusted to 8.0–9.0 using carbonate-bicarbonate buffer [0.2 M anhydrous sodium carbonate, 0.2 M sodium bicarbonate]) and sealed. Images of LSS staining, along with images of nuclei stained with DAPI, were segmented using the pretrained Cellpose model "cyto3" (Cellpose v. 3.0.7) (Stringer et al, 2021). Objects that touched the border of the image and objects below and above the set area thresholds (5k–100k pixels) were excluded from further analysis. The fluorescence signal of LSS was measured across the entire cell regions and background-corrected by subtracting the median signal of the entire image. For mTOR/LAMP2 and TSC2/LAMP2 staining, cells were blocked with Intercept Blocking Buffer (Licor, cat# 927-70001) diluted 1:1 in PBS for 1 h at room temperature. Cells were incubated overnight at 4 °C

in primary antibodies: TSC2 (CST, cat# 4308, 1:1250) or mTOR (CST, cat# 2938, 1:200) and LAMP2 (Santa Cruz, cat# sc18822, 1:100). Cells were then washed with PBS, incubated with secondary antibodies conjugated to Alexa Fluor 488 (Thermo Fisher Scientific, cat# A-11001, 1:1000) and Cy3 (Jackson ImmunoResearch, cat# 111-165-144, 1:1000) for 1 h at room temperature, washed with PBS again, and mounted on slides with Fluoromount G (SouthernBiotech, cat# 0100-01) for imaging using a Zeiss LSM 900 confocal microscope. TSC2:LAMP2 and mTOR:LAMP2 colocalization was quantified using the Zeiss ZenBlue 3.3 analysis suite.

## Filipin staining and analysis

Cells were cultured on coverslips and fixed with 4% paraformaldehyde. Cells were incubated with 0.1 mg/ml filipin III (Cayman Chemical, cat# 70440) for 45–60 min. Coverslips were washed twice with PBS, mounted, and sealed. To minimize bias, imaging analysis was performed in a blinded manner. Images of filipin staining were segmented using the pretrained Cellpose model "cyto3" (Cellpose v. 3.0.7) (Stringer et al, 2021). Objects that touched the border of the image and objects below and above the set area thresholds (25k–100k pixels) were excluded from further analysis. The fluorescence signal of filipin was measured across the entire cell regions with flat background correction (200 a.u.).

## RNA isolation, sequencing, and analysis

Total RNA was extracted from cells with Trizol (Ambion, cat# 15596018) and DNase treated, cleaned and concentrated using Zymo columns (Zymo Research, cat# R1013) following the manufacturer's instructions. RNA integrity number (RIN) was measured using the BioAnalyzer (Agilent Technologies) RNA 6000 Nano Kit to confirm RIN above 7 for each sample. The cDNA libraries, next-generation sequencing, and bioinformatics analysis was performed by Novogene. Raw and processed RNA-Seq data can be found on GEO (GSE291421).

## Immunoprecipitation

Cells were washed twice with cold 1x PBS. Cell lysates were collected in 1 ml RIPA lysis buffer (50 mm Tris-HCl (pH 7.4), 150 mm NaCl, 0.5% Triton X-100, 0.5% sodium deoxycholate, 2 mm $Na_3VO_4$, 2 mm NaF, the EDTA-free protease inhibitor cocktail (Sigma Aldrich, cat# 11836170001), phosphatase inhibitor cocktail 2 (Sigma Aldrich, cat# P5726) and phosphatase inhibitor cocktail 3 (Sigma Aldrich, cat# P0044). Cell lysates were incubated on ice for 5 min, pipetted up and down ten times, and then rotated on a rotatory shaker at 100 rpm for 60 min at 4 °C. Cell lysates were sonicated for 15 s on ice. Cell lysates were centrifuged at 14,000 rpm for 5 min at 4 °C. The supernatant was transferred to a new tube. Proteins were quantified using the BCA protein quantification method (Thermo Fisher Scientific, cat# 23225). Meanwhile, Pure-Proteome™ Protein G Magnetic Bead System (Sigma Aldrich, cat# LSKMAGG10) were precleaned three times using RIPA lysis buffer. Precleaned magnetic beads were conjugated either with IgG antibody (CST, cat# 2729S) or with LSS antibody (Sigma Aldrich, cat# HPA032060) at a concentration of 1 ug of antibody for 200 ug of protein by rotating them on a rotatory shaker at 100 rpm for 60 min at 4 °C. Additionally, protein samples were precleaned by

adding 15 ul of precleaned magnetic Beads to protein lysates and rotated them on a rotatory shaker at 100 rpm for 60 min at 4 °C. Magnetic Beads were pelleted by centrifuging the protein samples at 14,000 rpm for 5 min. Importantly, 10% of the total sample was collected as input (a loading control). Antibody conjugated beads were added to protein samples to immunoprecipitate LSS by rotating them on rotatory shaker at 100 rpm for overnight at 4 °C. Post-overnight incubation, magnetic Beads were washed two times using RIPA lysis buffer by rotating them on rotatory shaker at 100 rpm for 15 min at 4 °C. Post-washing, magnetic beads were incubated with 1x sample buffer on rotating heating block at 65 °C for 10 min at 1000 rpm. Samples were centrifuged at 14,000 rpm for 5 min at 4 °C and processed for immunoblotting.

### Hydroxyurea treatment

Cells were seeded at an equal density on plates or coverslips. Cells were treated with 0.5 mM Hydroxyurea (Sigma Aldrich, cat# H8627), to induce the replication stress, either in the presence or absence of AZD6738 (63 nM; ATR inhibitor; Selleckchem cat# S7693) for 24 h. Post-24h, cells were washed once with 1x PBS and fixed in 4% paraformaldehyde and then cells processed for filipin staining as described below.

### Quantification and statistical analysis

GraphPad Prism (version 10.0) was used to perform statistical analysis. Point estimates with standard deviations are reported, and the appropriate statistical test was performed using all observed experimental data. All statistical tests performed were two-sided, and $p$ values $< 0.05$ were considered statistically significant.

## Data availability

The mass spectrometry proteomics data have been deposited in the ProteomeXchange Consortium via the PRIDE partner repository with the dataset identifier PXD061265. The RNA-Seq data have been deposited in GEO (GSE291421).

The source data of this paper are collected in the following database record: biostudies:S-SCDT-10_1038-S44319-025-00451-3.

## Peer review information

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

## Acknowledgements

This work was supported by grants from the National Institutes of Health (R37CA240625 to KMA, R01CA259111 to KMA and NWS, R35GM155379 to AJV, T32GM133332 to ARC, and S10OD023402 and S10OD032141 to SLG), the Dept of Defense (HT9425-23-1-0288 to AJV and HT9425-24-1-0389 to NKT), the Melanoma Research Foundation (to RBG), the Ludwig Institute for Cancer Research—Princeton Branch (to AJV), the Ovarian Cancer Research Alliance (MIG-2023-2-1018 to AU), HERA Ovarian Cancer Foundation (to NKT, AU, and AA). This project has been made possible in part by grant number 2023-329680 (to KMK) from the Chan Zuckerberg Initiative DAF, an advised fund of Silicon Valley Community Foundation. Research reported in this publication was supported by the National Cancer Institute of the National Institutes of Health under Award Number P30CA047904. Synopsis figure created in BioRender. Aird, K. (2025) https://BioRender.com/ixqwwgc.

## Author contributions

Naveen Kumar Tangudu: Investigation; Visualization; Methodology; Writing—review and editing. Alexandra N Grumet: Investigation; Visualization; Methodology; Writing—review and editing. Richard Fang: Investigation; Writing—review and editing. Raquel Buj: Investigation; Visualization; Writing—review and editing. Aidan R Cole: Investigation; Writing—review and editing. Apoorva Uboveja: Investigation; Writing—review and editing. Amandine Amalric: Investigation; Writing—review and editing. Baixue Yang: Investigation. Zhentai Huang: Investigation. Cassandra Happe: Investigation. Mai Sun: Investigation. Stacy L Gelhaus: Supervision; Funding acquisition; Methodology. Matthew L MacDonald: Supervision; Funding acquisition; Methodology. Nadine Hempel: Writing—review and editing. Nathaniel W Snyder: Funding acquisition; Writing—review and editing. Katarzyna M Kedziora: Investigation; Methodology. Alexander J Valvezan: Conceptualization; Supervision; Visualization; Writing—original draft; Project administration; Writing—review and editing. Katherine M Aird: Conceptualization; Supervision; Funding acquisition; Visualization; Writing—original draft; Project administration; Writing—review and editing.

Source data underlying figure panels in this paper may have individual authorship assigned. Where available, figure panel/source data authorship is listed in the following database record: biostudies:S-SCDT-10_1038-S44319-025-00451-3.

## Disclosure and competing interests statement

The authors declare no competing interests.

# Expanded View Figures

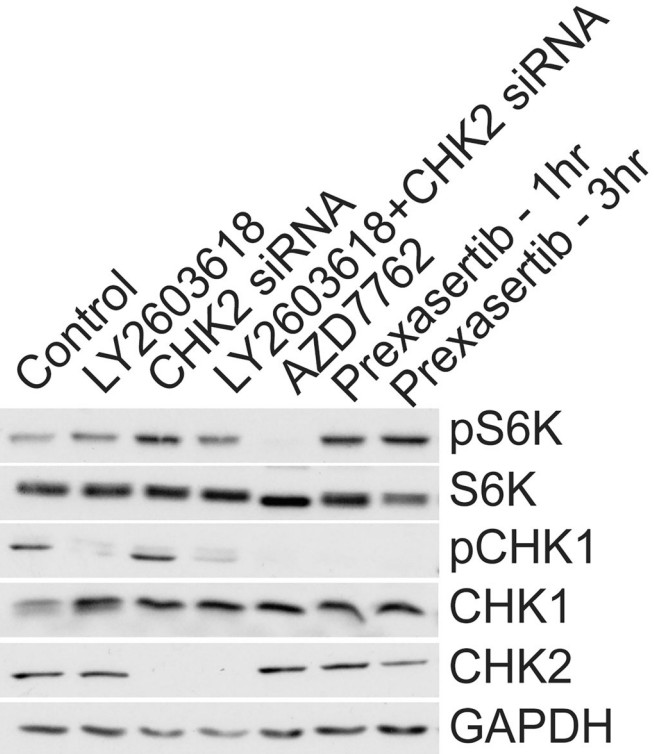

**Figure EV1. Suppression of CHK1 and CHK2 do not markedly decrease mTORC1 activity.**

HeLa cells were treated with the CHK1 inhibitor LY2603618 (2 μM, 3 h), CHK2 siRNA (50 nM, 48 h), the dual CHK1/CHK2 inhibitor AZD7762 (2 μM, 3 h), or the dual CHK1/CHK2 inhibitor Prexasertib [100 nM, 1 (lane 6) or 3 (lane 7) hrs], and the indicated proteins were assessed by western blotting. GAPDH was used as a loading control. Data information: Western blot data are representative of data from at least three independent experiments.

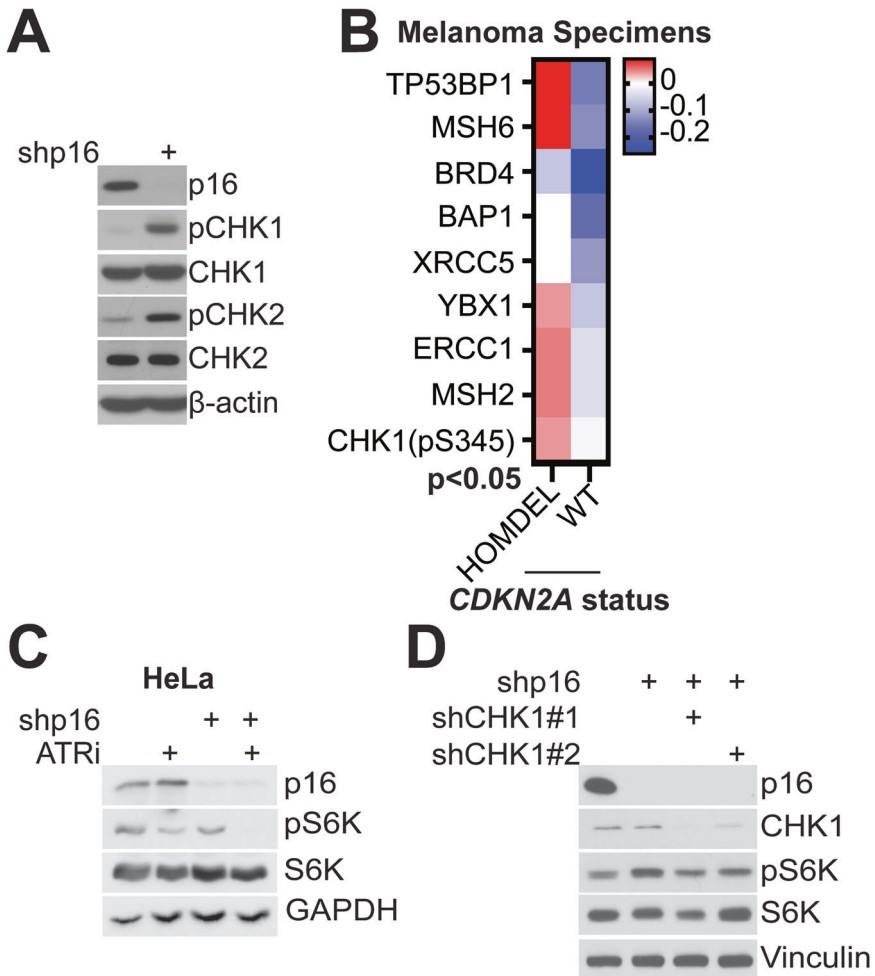

**Figure EV2. p16 knockdown increases activation of the ATR/ATM pathways; ATR-mediated mTORC1 activation in p16 knockdown cells is independent of CHK1.**

(A) SKMEL28 cells were transduced with a lentivirus expressing a short hairpin targeting GFP as a control or p16 (shp16), and the indicated proteins were assessed by western blotting. β-actin was used as loading controls. (B) RPPA results from Melanoma patient samples show upregulation of proteins related to the DNA damage response and repair in tumors with homozygous deletion (HOMDEL) of *CDKN2A* (encoding p16) compared to wildtype (WT) *CDKN2A* (all proteins $p < 0.05$). (C) HeLa cells were transduced with a lentivirus expressing a short hairpin targeting GFP as a control or p16 (shp16) and treated with 0.5 µM AZD6738 (ATRi) for 30 min. The indicated proteins were assessed by western blotting. GAPDH was used as a loading control. (D) Same as (A), but shp16 cells were transduced with a lentivirus expressing a short hairpin targeting GFP or two short hairpins targeting CHK1 (shCHK1 #1 and #2), and the indicated proteins were assessed by western blotting. Vinculin was used as a loading control. Data information: Western blot data in (A, C, D) are representative data from at least three independent experiments.

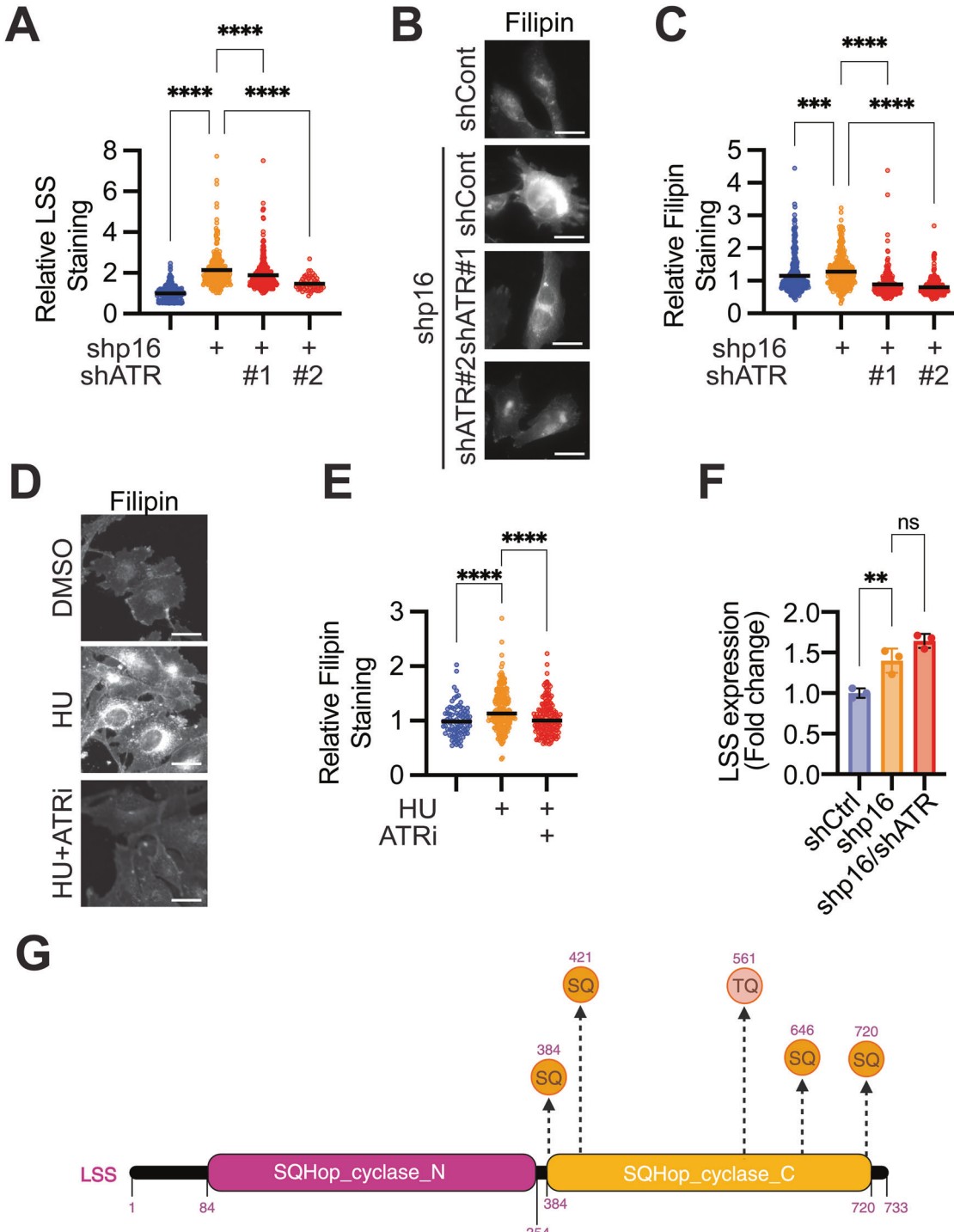

**Figure EV3.  LSS and cholesterol are downstream of ATR.**

(A–C) RPMI-7951 cells were transduced with lentivirus expressing shRNA targeting GFP as a control or p16 (shp16) with or without lentivirus expressing shRNA targeting ATR (shATR #1 and #2). (A) LSS expression was assessed by immunofluorescence staining and quantified. The graph represents individual normalized values and the mean. ****$p < 0.0001$. (B, C) Cholesterol abundance was assessed by filipin staining (B) and quantified (C). The graph represents individual normalized values and the mean. Scale bar $= 20\,\mu m$. ***$p = 0.0007$, ****$p < 0.0001$. (D, E) SKMEL28 control cells were treated with hydroxyurea (HU, 500 μM for 24 h) in the presence or absence of the ATRi (AZD6738, 63 nM for 24 h). Cholesterol abundance was assessed by filipin staining (D) and quantified (E). The graph represents individual normalized values and the mean. Scale bar $= 20\,\mu m$. ****$p < 0.0001$. (F) Fold change of *LSS* mRNA expression in the indicated groups. The graph represents mean ± SD. ns not significant, **$p = 0.007$. (G) Schematic of LSS protein with potential ATR phosphorylation sites. Data information: Data in (A–E) are representative data from one of three independent experiments. Graphs in (A, C, D) represent individual values and the mean. Data in (F) represent one experiment with three technical replicates. Statistical analysis in (A, C, E, F) was performed using one-way ANOVA with Šídák's multiple comparisons test.

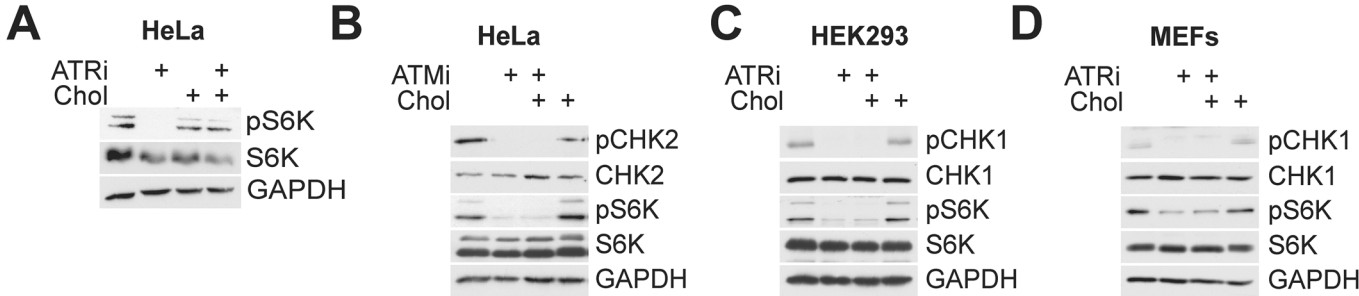

**Figure EV4. LSS and cholesterol are downstream of ATR.**

(A, B) shp16 HeLa cells were treated with ATRi (A) or ATMi (B) for 30 min in the presence or absence of supplementation with 50 μM cholesterol. (C, D) HEK293 cells (C) or MEFs (D) were treated vehicle or 0.5 μM AZD6738 (ATRi) for 30 min in the presence or absence of supplementation with 50 μM cholesterol. Data information: Western blot data in (A–D) are representative data from at least three independent experiments.

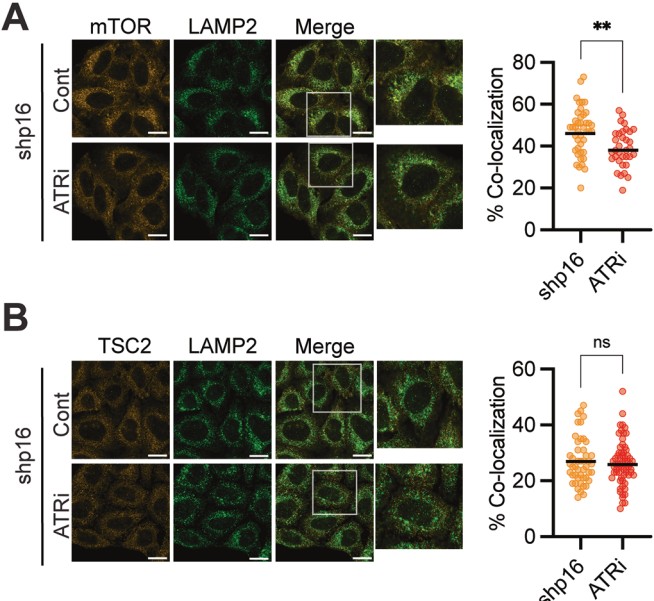

**Figure EV5. ATR inhibition decreases mTOR at the lysosome but has no effect on TSC2.**

(A, B) shp16 HeLa cells were treated for 30 min with 0.5 μM AZD6738 ATR inhibitor (ATRi) or vehicle. (A) Representative images of mTOR and LAMP2 immunofluorescence staining (left), which is quantified on the right. Scale bar = 20 μm. **p = 0.0015. (B) Representative images of TSC2 and LAMP2 immunofluorescence staining (left), which is quantified on the right. Scale bar = 20 μm. Scale bar = 20 μm. ns not significant. Data information: Representative data from one of three independent experiments is shown. Graphs represent individual normalized values and the mean. Statistical analysis in (A, B) was performed using an unpaired two-sided Student's *t*-test. **p = 0.0015, ns not significant.

