## [Peer Review File · EMBO Reports]

ATR promotes mTORC1 activity via de novo cholesterol synthesis

Naveen Tangudu, Alexandra Grumet, Richard Fang, Raquel Buj, Aidan Cole, Apoorva Uboveja, Amandine Amalric, Baixue Yang, Zhentai Huang, Cassandra Happe, Mai Sun, Stacy Gelhaus, Matthew MacDonald, Nadine Hempel, Nathaniel Snyder, Katarzyna Kedziora, Alexander Valvezan, and Katherine Aird

Corresponding author(s): Katherine Aird (kaa140@pitt.edu) , Alexander Valvezan (valvezan@cabm.rutgers.edu)

Review Timeline:

Submission Date:	5th Nov 24
Editorial Decision:	29th Nov 24
Revision Received:	14th Mar 25
Editorial Decision:	26th Mar 25
Revision Received:	1st Apr 25
Accepted:	7th Apr 25

Editor: Esther Schnapp

Transaction Report:

Dear Dr. Aird,

Thank you for the transfer of your manuscript to EMBO reports. We have now received the enclosed comments from 2 referees and given that both are in fair agreement I am making a decision on your ms now in order to save time.

As you will see, the referees acknowledge that the findings are interesting. However, they also have several suggestions for how your study could be strengthened and we think that all suggestions are good. While may be not all suggestions will need to be addressed, we do think that some further insight/data on how ATR impacts on mTORC1 activation will be required for the publication of your study here. I am happy to discuss a revision plan with you, either by email if you send me a proposed revision plan, or also a video chat would be fine. As you prefer. Please let me know how you would like to proceed.

I would thus like to invite you to revise your manuscript with the understanding that the referee concerns must be fully addressed and their suggestions taken on board. Please address all referee concerns in a complete point-by-point response. Acceptance of the manuscript will depend on a positive outcome of a second round of review. It is EMBO reports policy to allow a single round of major revision only and acceptance or rejection of the manuscript will therefore depend on the completeness of your responses included in the next, final version of the manuscript.

We realize that it is difficult to revise to a specific deadline. In the interest of protecting the conceptual advance provided by the work, we recommend a revision within 3 months (1st Mar 2025). Please discuss the revision progress ahead of this time with the editor if you require more time to complete the revisions.

- 1) A data availability section providing access to data deposited in public databases is missing. If you have not deposited any data, please add a sentence to the data availability section that explains that.
- 2) Your manuscript contains statistics and error bars based on $n=2$. Please use scatter blots in these cases. No statistics should be calculated if $n=2$.

3) We replaced Supplementary Information with Expanded View (EV) Figures and Tables that are collapsible/expandable online. A maximum of 5 EV Figures can be typeset. EV Figures should be cited as 'Figure EV1, Figure EV2' etc... in the text and their respective legends should be included in the main text after the legends of regular figures.

5) a complete author checklist, which you can download from our author guidelines . Please insert information in the checklist that is also reflected in the manuscript. The completed author checklist will also be part of the RPF.

6) Please note that all corresponding authors are required to supply an ORCID ID for their name upon submission of a revised manuscript (). Please find instructions on how to link your ORCID ID to your account in our manuscript tracking system in our Author guidelines

7) Before submitting your revision, primary datasets produced in this study need to be deposited in an appropriate public database (see <https://www.embopress.org/page/journal/14693178/authorguide#datadeposition>). Please remember to provide a reviewer password if the datasets are not yet public. The accession numbers and database should be listed in a formal "Data Availability" section placed after Materials & Method (see also <https://www.embopress.org/page/journal/14693178/authorguide#datadeposition>). Please note that the Data Availability Section is restricted to new primary data that are part of this study. * Note - All links should resolve to a page where the data can be accessed. *
If your study has not produced novel datasets, please mention this fact in the Data Availability Section.

- the name of the statistical test used to generate error bars and P values,
- the number (n) of independent experiments (please specify technical or biological replicates) underlying each data point,
- the nature of the bars and error bars (s.d., s.e.m.),
- If the data are obtained from n {less than or equal to} 2, use scatter blots showing the individual data points.

12) All Materials and Methods need to be described in the main text using our 'Structured Methods' format, which is required for all research articles. According to this format, the Methods section includes a Reagents and Tools Table (listing key reagents, experimental models, software and relevant equipment and including their sources and relevant identifiers) followed by a Methods and Protocols section describing the methods using a step-by-step protocol format. The aim is to facilitate adoption of the methodologies across labs. More information on how to adhere to this format as well as a downloadable template (.docx) for the Reagents and Tools Table can be found in our author guidelines:
<https://www.embopress.org/page/journal/14693178/authorguide#structuredmethods>.

An example of a Method paper with Structured Methods can be found here: <https://www.embopress.org/doi/full/10.1038/s44320-024-00037-6#sec-4>

I look forward to seeing a revised form of your manuscript when it is ready.

Referee #1:

This study provides an intriguing look at how ATR regulates the mTORC1 pathway through cholesterol metabolism, bridging the DDR and cellular growth signaling. While the role of ATR in influencing mTORC1 under stress is known, this work highlights its significance in non-stressed conditions, particularly through cholesterol synthesis driven by LSS. The study reveals that ATR promotes mTORC1 activation independently of CHK1 and the TSC complex, with supplementation of cholesterol or lanosterol restoring mTORC1 activity after ATR inhibition. By facilitating mTOR localization to lysosomes, ATR underscores a novel metabolic mechanism with potential implications for cancer therapy and age-related processes like T-cell proliferation. The manuscript is well-written, and its conclusions are generally well-supported by the data. However, a few points warrant further attention:

The connection between ATR and LSS is compelling but lacks mechanistic detail. It would be valuable to investigate whether ATR directly modulates LSS, perhaps via phosphorylation or transcriptional regulation.

The study heavily focuses on p16-deficient cells. Extending experiments to other cell types, such as immune or epithelial cells, or models with different genetic backgrounds, could establish whether the impact of ATR on cholesterol metabolism and mTORC1 is a generalizable phenomenon.

Given the emerging role of ATR in the cytoplasm, it would be worthwhile to test whether it interacts directly with lysosomal proteins such as LAMP2 or NPC1, which could add depth to the proposed mechanism.

Since DDR and mTORC1 are closely linked, exploring whether DNA damage influences the effect of ATR on cholesterol synthesis could provide deeper insights into the bidirectional relationship between these pathways.

Referee #2:

Tangudu and colleagues present an exciting connection between ATR and mTORC1 signaling, revealing cholesterol metabolism as a key intermediary. Their findings demonstrate that ATR promotes de novo cholesterol synthesis and mTORC1 activation by upregulating lanosterol synthase (LSS), independently of CHK1 and the TSC complex. Remarkably, the inhibition of ATR reduced mTORC1 activity, a defect rescued by lanosterol or cholesterol supplementation, emphasizing cholesterol's role in mTOR lysosomal localization and activation.

This study offers an exciting insight into how ATR and mTORC1 signaling are linked via cholesterol metabolism.

1) One fundamental point would be to address the molecular mechanisms underlying the regulation of LSS by ATR. Does ATR act through established DNA damage response (DDR) effectors, such as transcription factors like E2F1, or does it engage previously unrecognized regulatory pathways? Additionally, while ATR's potential cytoplasmic activity is noted, the possibility that ATR directly phosphorylates LSS to influence its stability or enzymatic function remains unexplored. Experimental validation of these mechanisms would significantly enhance our understanding of how ATR integrates DDR signaling with metabolic regulation.

2) While this study highlights cholesterol's role in mTORC1 activation, its broader implications in tumorigenesis warrant further discussion. Cholesterol also supports membrane synthesis in proliferating cells and contributes to steroid biosynthesis, both of which are crucial in cancer progression. Expanding the discussion to encompass these roles would highlight the ATR-LSS axis's multifaceted contributions to cancer biology.

3) Given prior work showing LYCHOS as a cholesterol-sensing mediator for mTORC1, examining whether LYCHOS is involved downstream of ATR could clarify how cholesterol signals are transduced to mTORC1.

4) ATR inhibitors are under clinical development, yet this study suggests potential metabolic side effects related to mTORC1 signaling and cholesterol synthesis. Expanding on the implications for therapy design, including combinatorial strategies to mitigate side effects, would add translational value.

5) While ATR is highlighted, ATM inhibition also impacts mTORC1, albeit less strongly. A more detailed exploration of the differential and overlapping roles of ATR and ATM in cholesterol metabolism and mTORC1 regulation could provide a more comprehensive understanding of these pathways and their interplay.

We would like to express our sincere gratitude to the referees and the Editor for their constructive and thoughtful review of our review submitted to *EMBO Reports*. We are grateful for the acknowledgment from the Editor and all Reviewers regarding the significance and impact of our research, including “***the findings are interesting***” (Senior Editor); “***This study provides an intriguing look at how ATR regulates the mTORC1 pathway through cholesterol metabolism***” (Reviewer #1); “***This study offers an exciting insight into how ATR and mTORC1 signaling are linked via cholesterol metabolism***” (Reviewer #2). This positive evaluation validates the scientific merit of our work in investigating that ATR promotes mTORC1 activity via *de novo* cholesterol synthesis.

We truly appreciate the critical role and effort of this set of knowledgeable and helpful reviewers. Our commitment to addressing the Reviewers' comments is reflected in the comprehensive point-by-point response outlined below and a **significant amount of new data presented in the revised manuscript**. We believe that these revisions have resulted in a more compelling, robust, cohesive, and scientifically sound manuscript connecting ATR and mTORC1 signaling.

Reviewer #1

Comment 1: The connection between ATR and LSS is compelling but lacks mechanistic detail. It would be valuable to investigate whether ATR directly modulates LSS, perhaps via phosphorylation or transcriptional regulation.

Response: We thank the reviewer for this thoughtful suggestion to deepen the mechanistic insights of our work. We have provided new experimental evidence that ATR directly modulates LSS through phosphorylation but not transcription, which we will detail below.

Regulation of LSS by ATR-mediated phosphorylation: LSS has 5 potential ATR phospho sites (S*/T*Q) (**Fig. S3F**). Unfortunately, there are no commercially available LSS phospho-specific antibodies. Therefore, to investigate whether ATR directly modulates LSS through phosphorylation, we immunoprecipitated LSS in p16 knockdown cells with or without ATR knockdown and assessed potential phosphorylation sites via immunoblotting using a Phospho-ATM/ATR Substrate antibody. We found that ATR knockdown reduces phosphorylation of LSS (**new Fig. 3H**). These data point to LSS as a direct ATR substrate, which has been noted in the revised text (**Page 9-10, Lines 212-214**). Further mechanistic studies to determine which residue(s) ATR phosphorylates and whether/how they affect LSS levels, activity, or localization will be interesting for future studies. We have discussed this in the revised text (**Page 13, 272-278**).

Transcriptional Regulation by ATR: To evaluate whether ATR directly modulates LSS through transcriptional regulation, we performed RNA-Seq on p16 knockdown cells with or without ATR knockdown. We observed a modest (<1.5 fold) increase in LSS expression upon knockdown of p16; however, ATR knockdown did not rescue this effect (**new Fig. S3E**). These data demonstrate that ATR does not transcriptionally regulate LSS.

Comment 2: The study heavily focuses on p16-deficient cells. Extending experiments to other cell types, such as immune or epithelial cells, or models with different genetic backgrounds, could establish whether the impact of ATR on cholesterol metabolism and mTORC1 is a generalizable phenomenon.

Response: We appreciate the reviewer's thoughtful suggestion. In addition to the p16-deficient melanoma cells, we also found that ATR regulates mTORC1 activation in HeLa cells (a human epithelial adenocarcinoma cell line), HEK293 cells (Human Embryonic Kidney cells, an epithelial cell line), and MEFs (Mouse Embryonic Fibroblasts) (**Fig. 1A-F, EV1, and EV2C**), suggesting that the mechanism of ATR mediating mTORC1 activity is a generalizable phenomenon. To better understand whether this is mediated via cholesterol in all cell lines, we inhibited ATR in these cell lines and supplemented them with cholesterol and assessed mTORC1 activation via S6K phosphorylation. Unexpectedly, we found that cholesterol rescues pS6K in both of our cancer cell models (**new Fig. 4A-B, 4E, 4G and Fig. EV4A-B**) while cholesterol did not rescue pS6K in normal cells (**new Fig. EV4C-D**). Together, these data have uncovered a surprising finding that ATR can regulate mTORC1 activity via cholesterol in cancer cells, whereas normal cells may have an alternative pathway. Although outside the scope of the current study, it is intriguing to speculate that this is due to lipid metabolic derangement or high basal replication stress of cancer cells. It is also possible that these cells take up less cholesterol. We have noted that this phenomenon seems to be generalizable to cancer but not normal cells (**Page 10, Lines 225-229**) and expanded on this interesting observation in the revised Discussion (**Page 13, Lines 282-286**).

Comment 3: Given the emerging role of ATR in the cytoplasm, it would be worthwhile to test whether it interacts directly with lysosomal proteins such as LAMP2 or NPC1, which could add depth to the proposed mechanism.

Response: We appreciate the reviewer for this intriguing comment regarding ATR in the cytoplasm. We found that ATR is upstream of LSS, which is not known to be at the lysosome. Interestingly, prior high throughput proteomics studies found LSS in or near the nucleus (MacDonald et al., 2024; Wang et al., 2017). Together, these data indicate it is likely that the ATR and LSS interaction does not occur at the lysosome. Moreover, we agree with the reviewer that further assessment of ATR and LSS localization would provide valuable information regarding the mechanism and may also point to why this mechanism is cancer cell specific. We did find that LSS is phosphorylated by ATR (**new Fig. 3H**; an experiment in response to Comment #1), and future studies will be aimed at probing whether this occurs in the nucleus. We have expanded on this idea in the revised Discussion (**Page 13, Lines 272-276**).

Comment 4: Since DDR and mTORC1 are closely linked, exploring whether DNA damage influences the effect of ATR on cholesterol synthesis could provide deeper insights into the bidirectional relationship between these pathways.

Response: We thank the reviewer for this suggestion. To explore whether DNA damage influences the effect of ATR on cholesterol synthesis, we induced replication stress in cancer cells by treating with hydroxyurea to activate ATR and quantified cholesterol using filipin staining. Indeed, inducing replication stress in cancer cells increased cholesterol levels, which was abrogated by ATR inhibition (**new Fig. EV3D-E**). These data provide important insights into the bidirectional relationship between the DNA damage and mTORC1 pathways. We also believe these data may provide additional evidence to explain why cancer cells, which have high basal levels of replication stress, have a more active ATR-cholesterol-mTORC1 signaling pathway than normal cells. This point has been added to the revised text (**Page 13, Lines 282-286**).

Reviewer #2

Comment 1: One fundamental point would be to address the molecular mechanisms underlying the regulation of LSS by ATR. **A)** Does ATR act through established DNA damage response (DDR) effectors, such as transcription factors like E2F1, or does it engage previously unrecognized regulatory pathways? **B)** Additionally, while ATR's potential cytoplasmic activity is noted, the possibility that ATR directly phosphorylates LSS to influence its stability or enzymatic function remains unexplored. Experimental validation of these mechanisms would significantly enhance our understanding of how ATR integrates DDR signaling with metabolic regulation.

Response: We thank the reviewer for this insightful suggestion. **Reviewer #1** had a similar question regarding the transcriptional regulation of *LSS* by ATR (Comment #1 above). Using RNA-Seq, we found that ATR does not affect *LSS* mRNA expression (**new Fig. EV4F**), demonstrating that ATR does not upregulate *LSS* through a transcriptional factor like E2F1.

The second question regarding whether ATR directly phosphorylates *LSS* is an intriguing one. *LSS* has 5 potential ATR phospho sites (S*/T*Q) (**Fig. EV4G**). Unfortunately, there are no commercially available *LSS* phospho-specific antibodies. Therefore, to investigate whether ATR directly modulates *LSS* through phosphorylation, we immunoprecipitated *LSS* in p16 knockdown cells with or without ATR knockdown and assessed potential phosphorylation sites via immunoblotting using a Phospho-ATM/ATR Substrate antibody. We found that ATR knockdown reduces phosphorylation of *LSS* (**new Fig. 3H**). These data point to *LSS* as a direct ATR substrate, which has been noted in the revised text (**Pages 9-10, Lines 212-215**). How this phosphorylation affects *LSS* remains to be explored. Further mechanistic studies to determine the residue(s) ATR phosphorylates and how that event directly affects *LSS* stability, activity, or localization will be interesting for future studies. We have discussed this in the revised text (**Page 13, Lines 272-276**).

Comment 2: While this study highlights cholesterol's role in mTORC1 activation, its broader implications in tumorigenesis warrant further discussion. Cholesterol also supports membrane synthesis in proliferating cells and contributes to steroid biosynthesis, both of which are crucial in cancer progression. Expanding the discussion to encompass these roles would highlight the ATR-LSS axis's multifaceted contributions to cancer biology.

Response: We thank reviewer for this comment. We have discussed these points in the manuscript and expanded the revised text by adding more discussion points to emphasize the ATR-LSS axis's multifaceted contributions to cancer biology, including tumor progression, tumor microenvironment reprogramming, and metastasis (**Page 14, Lines 296-306**).

Comment 3: Given prior work showing LYCHOS as a cholesterol-sensing mediator for mTORC1, examining whether LYCHOS is involved downstream of ATR could clarify how cholesterol signals are transduced to mTORC1.

Response: We thank reviewer for this suggestion. Unfortunately, due to unanticipated technical challenges, we have been unable to knockdown LYCHOS in our cell lines. In fact, we attempted this knock down experiment four independent times with 5 independent shRNAs. To our knowledge, LYCHOS is the only cholesterol sensor at the lysosome (Shin et al., 2022). Therefore, given that we observed increased mTOR at the lysosome in the context of high cholesterol (**Fig. 5**), it is probable that the cholesterol generated downstream of ATR is sensed by LYCHOS to activate mTORC1. We have clarified this point in the revised text (**Page 13-14, Lines 291-296**).

Comment 4: ATR inhibitors are under clinical development, yet this study suggests potential metabolic side effects related to mTORC1 signaling and cholesterol synthesis. Expanding on the implications for therapy design, including combinatorial strategies to mitigate side effects, would add translational value.

Response: We thank reviewer for this helpful suggestion. We expanded our discussion to address this thoughtful comment from the reviewer on implications of reduced mTORC1 activity downstream of ATR inhibitors, effects of cholesterol reduction on tumor growth and the tumor microenvironment, effects on the immune system and response to immunotherapies, and implications for treatment regimens and mitigating potential toxicities. We further discussed these points in the revised discussion section (**Pages 14-15, Lines 308-323**).

Comment 5: While ATR is highlighted, ATM inhibition also impacts mTORC1, albeit less strongly. A more detailed exploration of the differential and overlapping roles of ATR and ATM in cholesterol metabolism and mTORC1 regulation could provide a more comprehensive understanding of these pathways and their interplay.

Response: We thank reviewer for this intriguing comment. As the reviewer pointed out, in **Fig. 1 and 2D** we have shown that ATM knockdown and inhibition impacts mTORC1, albeit less strongly. To explore the role of ATM in cholesterol metabolism and mTORC1 regulation, we inhibited ATM in both control and TSC2 deficient cells and supplemented with cholesterol to assess mTORC1 activation via pS6K expression. We found that unlike ATR, which mediates mTORC1 via cholesterol in cancer cell lines, and independent of the TSC complex (**Fig. 1E, 3-4 and EV4A-B**), ATM mediates mTORC1 via the TSC complex (**Fig. 1G**), and ATM inhibition is not rescued by cholesterol even in cancer cells (**Fig. EV4B**). Together, our data demonstrate that ATR and ATMs have differential roles in cholesterol metabolism and mTORC1 regulation.

References

MacDonald, K.M., Khan, S., Lin, B., Hurren, R., Schimmer, A.D., Kislinger, T., and Harding, S.M. (2024). The proteomic landscape of genotoxic stress-induced micronuclei. *Molecular cell* 84, 1377-1391.e1376.

Shin, H.R., Citron, Y.R., Wang, L., Tribouillard, L., Goul, C.S., Stipp, R., Sugasawa, Y., Jain, A., Samson, N., Lim, C.Y., *et al.* (2022). Lysosomal GPCR-like protein LYCHOS signals cholesterol sufficiency to mTORC1. *Science (New York, NY)* 377, 1290-1298.

Wang, J., Mauvoisin, D., Martin, E., Atger, F., Galindo, A.N., Dayon, L., Sizzano, F., Palini, A., Kussmann, M., Waridel, P., *et al.* (2017). Nuclear Proteomics Uncovers Diurnal Regulatory Landscapes in Mouse Liver. *Cell metabolism* 25, 102-117.

Dear Dr. Aird,

Thank you for the submission of your revised manuscript. We have now received the enclosed reports from the referees and I am happy to say that both support its publication now. Only a few editorial requests will need to be addressed before we can proceed with the official acceptance of your manuscript:

- Please reduce the number of keywords to 5.
- Please correct the conflict of interest subheading to "Disclosure and Competing Interests Statement"
- There are 2 author name discrepancies: Naveen Kumar Tangudu in the ms vs. Naveen Tangudu in our online system; Katarzyna M. Kedziora in the ms vs. Kasia Kedziora in the online system. Please correct.
- Please remove the author credits from the ms file. All credits need to be entered during online ms submission.
- Some FUNDING INFO is missing in our online system: the Ludwig Institute for Cancer Research - Princeton Branch, please add when you submit the final ms.
- It is unclear what the file is that is uploaded as "Expanded View". Is this a Dataset? Please label correctly and add a title or legend. If it is a Dataset it needs to be called Dataset EV1.
- The Source Data (SD) needs to be uploaded as one SD file per main figure. SD for Fig 3C seems to be missing, or is this deposited to PRIDE PXD061265?
- The Data Availability Section (DAS), the Acknowledgments and the Disclosure Statement (in that order) need to be provided after the Methods and before the References. The DAS needs to provide a website link to your deposited data.
- Materials and Methods should be just Methods.
- Please provide the exact p values (as reasonable) in the legends of figures 3C, E, G; 5A, B; EV3 A, C, E, F; EV5 A, B.

I would like to suggest some minor changes to the abstract that needs to be written in present tense:

DNA damage and cellular metabolism exhibit a complex interplay characterized by bidirectional feedback. Key mediators of these pathways include Ataxia Telangiectasia and Rad3-related protein (ATR) and the mechanistic Target of Rapamycin Complex 1 (mTORC1), respectively. Previous studies established ATR as a regulatory upstream factor of mTORC1 during replication stress; however, the precise mechanisms remain poorly defined. Additionally, the activity of this signaling axis in unperturbed cells has not been extensively investigated. We demonstrate that ATR promotes mTORC1 activity across various cellular models under basal conditions. This effect is enhanced in cells following the loss of p16. Mechanistically, ATR promotes de novo cholesterol synthesis and mTORC1 activation through the phosphorylation and upregulation of lanosterol synthase (LSS), independently of both CHK1 and the TSC complex. Interestingly, this pathway is distinct from the regulation of mTORC1 by ATM and may be specific to cancer cells. Finally, ATR-mediated cholesterol increase correlates with enhanced localization of mTOR to lysosomes. Collectively, our findings demonstrate a novel connection linking ATR and mTORC1 signaling through the modulation of cholesterol metabolism.

I think it would be good to specify the cells used in your study a little more in the abstract. And to specify what p16 is.

EMBO press papers are accompanied online by A) a short (1-2 sentences) summary of the findings and their significance, B) 2-3 bullet points highlighting key results and C) a synopsis image that is exactly 550 pixels wide and 200-600 pixels high (the height is variable). The synopsis image should provide a sketch of the major findings, like a graphical abstract. Please note that text needs to be readable at the final size. Please send us this information along with the final manuscript.

Esther Schnapp, PhD
Senior Editor

EMBO reports

Referee #1:

The authors have adequately addressed the Reviewers' concerns/remarks.

Referee #2:

The authors have addressed my comments. I congratulate the authors on this excellent research article!

All editorial and formatting issues were resolved by the authors.

Katherine Aird
University of Pittsburgh School of Medicine
United States

Dear Dr. Aird,

I am very pleased to accept your manuscript for publication in the next available issue of EMBO reports. Thank you for your contribution to our journal.

Best,
Esther
